# Understanding Expressivity of GNN in Rule Learning

**Haiquan Qiu[1], Yongqi Zhang[2], Yong Li[1], Quanming Yao[1]***
[1]Department of Electronic Engineering, Tsinghua University
[2]The Hong Kong University of Science and Technology (Guangzhou)
`qyaoaa@tsinghua.edu.cn`

## Abstract

Rule learning is critical to improving knowledge graph (KG) reasoning due to their ability to provide logical and interpretable explanations. Recently, Graph Neural Networks (GNNs) with tail entity scoring achieve the state-of-the-art performance on KG reasoning. However, the theoretical understandings for these GNNs are either lacking or focusing on single-relational graphs, leaving what the kind of rules these GNNs can learn an open problem. We propose to fill the above gap in this paper. Specifically, GNNs with tail entity scoring are unified into a common framework. Then, we analyze their expressivity by formally describing the rule structures they can learn and theoretically demonstrating their superiority. These results further inspire us to propose a novel labeling strategy to learn more rules in KG reasoning. Experimental results are consistent with our theoretical findings and verify the effectiveness of our proposed method. The code is publicly available at `https://github.com/LARS-research/Rule-learning-expressivity`.

## 1 Introduction

A knowledge graph (KG) (Battaglia et al., 2018; Ji et al., 2021) is a type of graph where edges represent multiple types of relationships between entities. These relationships can be of different types, such as friend, spouse, coworker, or parent-child, and each type of relationship is represented by a separate edge. By encapsulating the interactions among entities, KGs provide a way for machines to understand and process complex information. KG reasoning refers to the task of deducing new facts from the existing facts in KG. This task is important because it helps in many real-world applications, such as recommendation systems (Cao et al., 2019) and drug discovery (Mohamed et al., 2019).

With the success of graph neural networks (GNNs) in modeling graph-structured data, GNNs have been developed for KG reasoning in recent years. Classical methods such as R-GCN (Schlichtkrull et al., 2018) and CompGCN (Vashishth et al., 2020) are proposed for KG reasoning by aggregating the representations of two end entities of a triplet. And they are known to fail to distinguish the structural role of different neighbors. GraIL (Teru et al., 2020) and RED-GNN (Zhang & Yao, 2022) tackle this problem by encoding the subgraph around the target triplet. GraIL predicts a new triplet using the subgraph representations, while RED-GNN employs dynamic programming for efficient subgraph encoding. Motivated by the effectiveness of heuristic metrics over paths between a link, NBFNet (Zhu et al., 2021) proposes a neural network based on Bellman-Ford algorithm for KG reasoning. AdaProp (Zhang et al., 2023) and A*Net (Zhu et al., 2022) enhance the scalability of RED-GNN and NBFNet respectively by selecting crucial nodes and edges iteratively. Among these methods, NBFNet, RED-GNN and their variants score a triplet with its tail entity representation and achieve state-of-the-art (SOTA) performance on KG reasoning. However, these methods are motivated by different heuristics, e.g., Bellman-Ford algorithm and enclosing subgraph encoding, which make the understanding of their effectiveness for KG reasoning difficult.

In this paper, inspired by the importance of rule learning in KG reasoning, we propose to study expressivity of SOTA GNNs for KG reasoning by analyzing the kind of rules they can learn. First, we unify SOTA GNNs for KG reasoning into a common framework called QL-GNN, based on the

---

*Quanming Yao is the corresponding author.

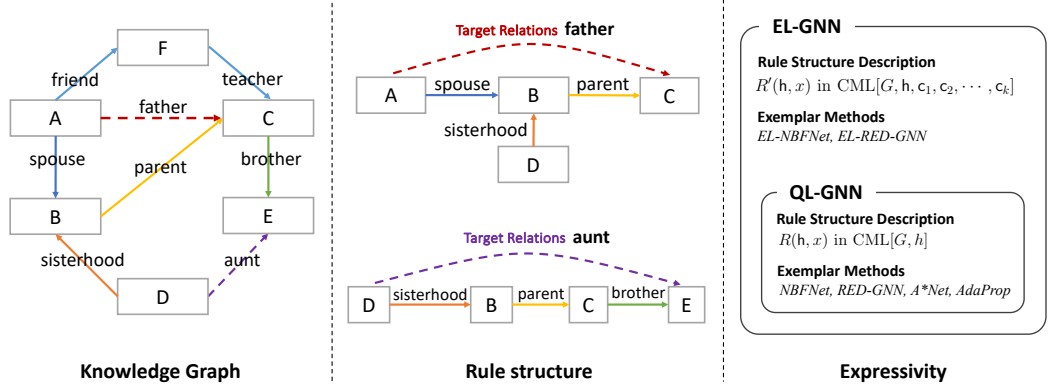

Figure 1: The existence of a triplet in KG is determined by the corresponding rule structure. We investigates the kind of rule structures can be learned by SOTA GNNs for KG reasoning (i.e., QL-GNN), and proposes EL-GNN, which can learn more rule structures compared to QL-GNN.

observation that they score a triplet with its tail entity representation and essentially extract rule structures from subgraphs with same pattern. Then, we analyze the logical expressivity of QL-GNN to study its ability of learning rule structures. The analysis helps us reveal the underneath theoretical reasons that contribute to the empirical success of QL-GNN, elucidating their effectiveness over classical methods. Specifically, our analysis is based on the formal description of rule structures in graph, which differs from previous analysis that relies on graph isomorphism testing (Xu et al., 2019; Zhang et al., 2021) and focuses on the expressivity of distinguishing various rules. The new analysis tool allows us to understand the rules learned by QL-GNN and reveals the maximum expressivity that QL-GNN can generalize through training. Based on the new theory, we also uncover the deficiencies of QL-GNN in learning rule structures and we propose EL-GNN based on labeling trick as an improvement upon QL-GNN to improve its learning ability. In summary, our paper has the following contributions:

- Our work unifies state-of-the-art GNNs for KG reasoning into a common framework named QL-GNN, and analyzes their logical expressivity to study their ability of learning rule structures, explaining their superior performance over classical methods.

- The logical expressivity of QL-GNN demonstrates its capability in learning a particular class of rule structures. Consequently, based on further theoretical analysis, we introduce EL-GNN, a novel GNN designed to learn rule structures that are beyond the learning capacity of QL-GNN.

- Synthetic datasets are generated to evaluate the expressivity of various GNNs, whose experimental results are consistent with our theory. Also, results of the proposed labeling method show improved performance on real datasets.

## 2 A COMMON FRAMEWORK FOR THE STATE-OF-THE-ART METHODS

To study the state-of-the-art GNNs for KG reasoning, we find that they (e.g., RED-GNN and NBFNet) essentially learn rule structures from GNN's tail entity representation which encodes subgraphs with the same pattern, i.e., a subgraph with the query entity as the source node and the tail entity as the sink node. Based on this observation, we are motivated to derive a common framework for these SOTA methods and analyze their ability of learning rule structures with the derived framework.

Given a query $(h, R, ?)$, the labeling trick of query entity $h$ ensures the SOTA methods to extract rules from a graph with the same pattern because it makes the query entity distinguishable among all entities in graph. Therefore, we unify NBFNet, RED-GNN and their variants to a common framework called Query Labeling (QL) GNN (see correspondence in Appdendix B). For a query $(h, R, ?)$, QL-GNN first applies labeling trick by assigning special initial representation $\mathbf{e}_h^{(0)}$ to entity $h$, which make the query entity distinguishable from other entities. Base on these initial features, QL-GNN aggregates

entity representations with a $L$-layer message passing neural network (MPNN) for each candidate $t \in \mathcal{V}$. MPNN's last layer representation of entity $t$ in QL-GNN is denoted as $\mathbf{e}_t^{(L)}[h]$ indicating its dependency on query entity $h$. Finally, QL-GNN scores new facts $(h, R, t)$ with tail entity representation $\mathbf{e}_t^{(L)}[h]$. For example, NBFNet uses the score function $s(h, R, t) = \text{FFN}(\mathbf{e}_t^{(L)}[h])$ for new triplet $(h, R, t)$ where $\text{FFN}(\cdot)$ denotes a feed-forward neural network.

Even RED-GNN, NBFNet and their variant may take the different MPNNs to calculate $\mathbf{e}_t^{(L)}[h]$, without loss of generality, their MPNNs can take the following form in QL-GNN (omit $[h]$ for simplicity):

$$\mathbf{e}_v^{(k)} = \delta\left(\mathbf{e}_v^{(k-1)}, \phi\left(\{\{\psi(\mathbf{e}_u^{(k-1)}, R) | u \in \mathcal{N}_R(v), R \in \mathcal{R}\}\}\right)\right), \tag{1}$$

where $\delta$ and $\phi$ are combination and aggregation functions respectively, $\psi$ is the message function encoding the relation $R$ and entity $u$ neighboring to $v$, $\{\{\cdots\}\}$ is a multiset, and $\mathcal{N}_R(v)$ is the neighboring entity set $\{u|(u, R, v) \in \mathcal{E}\}$.

## 3 EXPRESSIVITY OF QL-GNN

In this section, we explore the logical expressivity of QL-GNN to analyze the types of rule structures QL-GNN can learn. First, we provide the logic to describe rules in KGs. Then, we analyze logical expressivity of QL-GNN using Theorem 3.2 and Corollary 3.3, formally demonstrating the kind of rule structures it can learn. Finally, we compare QL-GNN with classical methods and highlight its superior expressivity in KG reasoning.

### 3.1 EXPRESSIVITY ANALYSIS WITH LOGIC OF RULE STRUCTURES

From previous works of rule mining on KG (Yang et al., 2017; Sadeghian et al., 2019), rule structures are usually described as a formula in first-order logic. We also follow this way to formally describe the rule structures in KG. Therefore, we have the following correspondence between the elements in rule structures and logic:

- Variable: variables denoted with lowercase italic letters $x, y, z$ represent entities in a KG;
- Unary predicate: unary predicate $P_i(x)$ is corresponding to the entity property $P_i$ in a KG, e.g., $\text{red}(x)$ denotes the color of an entity $x$ is red;
- Binary predicate: binary predicate $R_j(x, y)$ is corresponding to the relation $R_j$ in a KG, e.g., $\text{father}(x, y)$ denotes $x$ is the father of $y$;
- Constant: constant denoted with lowercase letters $\mathsf{h}, \mathsf{c}$ with serif typestyle is the unique identifier of some entity in a KG.

Except from the above elements, the quantifier $\exists$ expresses the existence of entities satisfying a condition, $\forall$ expresses universal quantification, and $\exists^{\geq N}$ represents the existence of at least $N$ entities satisfying a condition. The logical connective $\wedge$ denotes conjunction, $\vee$ denotes disjunction, and $\top$ and $\bot$ represent true and false, respectively. Using these symbols, rule structures can be represented by describing their elements directly. For example, $C_3(x, y) := \exists z_1 z_2, R_1(x, z_1) \wedge R_2(z_1, z_2) \wedge R_3(z_2, y)$ in Figure 2 describes a chain-like structure between $x$ and $y$ with three relations $R_1, R_2, R_3$. Rule structures can be represented using the rule formula $R(x, y)$, and the existence of a rule structure for the triplet $(h, R, t)$ is equivalent to the satisfaction of the rule formula $R(x, y)$ at the entity pair $(h, t)$. In this paper, **logical expressivity** of GNN is a measurement of the ability of GNN to learn logical formulas and is defined as the set of logical formulas that GNN can learn. Therefore, since rule structures can be described by logical formulas, the logical expressivity of QL-GNN can determine their ability to learn rule structures in KG reasoning.

### 3.2 WHAT KIND OF RULE STRUCTURES CAN QL-GNN LEARN?

In this section, we analyze the logical expressivity of QL-GNN regarding what kind of rule structure it can learn. Given a query $(h, R, ?)$, we first have the following proposition about the rule formula describing a rule structure.

**Proposition 3.1.** *The rule structure for query $(h, R, ?)$ can be described with rule formula $R(x, y)$ or rule formula $R(\mathsf{h}, x)$ [1] where $\mathsf{h}$ is the logical constant assigned to query entity $h$.*

QL-GNN applies labeling trick to the query entity $h$, which can be equivalently seen as assigning constant $\mathsf{h}$ to query entity $h$[2]. With Proposition 3.1 (proven in Appendix A), the logical expressivity of QL-GNN can be analyzed by the types of rule formula $R(\mathsf{h}, x)$ it can learn. In this case, the rule structure of triplet $(h, R, t)$ exists if and only if the logical formula $R(\mathsf{h}, x)$ is satisfied at entity $t$.

### 3.2.1 EXPRESSIVITY OF QL-GNN

Before presenting the logical expressivity of QL-GNN, we start by explaining how QL-GNN learns the rule formula $R(\mathsf{h}, x)$. Following the definition in Barceló et al. (2020), we treat $R(\mathsf{h}, x)$ as a binary classifier. When given a candidate tail entity $t$, if the triplet $(h, R, t)$ exists in a KG, the binary classifier $R(\mathsf{h}, x)$ should output true; otherwise, it should output false. If QL-GNN can learn the rule formula $R(\mathsf{h}, x)$, it implies that QL-GNN can estimate binary classifier $R(\mathsf{h}, x)$. Consequently, if the rule formula $R(\mathsf{h}, x)$ is satisfied at entity $t$, the representation $\mathbf{e}_t^{(L)}[h]$ is mapped to a high probability value, indicating the existence of triplet $(h, R, t)$ in KG. Conversely, when the rule formula is not satisfied at $t$, $\mathbf{e}_t^{(L)}[h]$ is mapped to a low probability value, indicating the absence of the triplet.

The rule structures that QL-GNN can learn are described by a family of logic called graded modal logic (CML) (De Rijke, 2000; Otto, 2019). CML is defined by recursion with the base elements $\top, \bot$, all unary predicates $P_i(x)$, and the recursion rule: if $\varphi(x), \varphi_1(x), \varphi_2(x)$ are formulas in CML, $\neg\varphi(x), \varphi_1(x) \land \varphi_2(x), \exists^{\geq N} y\, (R(y, x) \land \varphi(y))$ are also formulas in CML. Since QL-GNN introduces a constant $\mathsf{h}$ to the query entity $h$, we use the notation $\text{CML}[G, \mathsf{h}]$ to denote the CML recursively built from base elements in $G$ and constant $\mathsf{h}$ (equivalent to constant predicate $P_h(x)$). Then, the following theorem and corollary show the expressivity of QL-GNN for KG reasoning.

**Theorem 3.2** (Logical expressivity of QL-GNN). *For KG reasoning, given a query $(h, R, ?)$, a rule formula $R(\mathsf{h}, x)$ is learned by QL-GNN if and only if $R(\mathsf{h}, x)$ is a formula in $CML[G, \mathsf{h}]$.*

**Corollary 3.3.** *The rule structures learned by QL-GNN can be constructed with the recursion:*
- ***Base case:** all unary predicates $P_i(x)$ can be learned by QL-GNN; the constant predicate $P_h(x)$ can be learned by QL-GNN;*
- ***Recursion rule:** if the rule structures $R_1(\mathsf{h}, x), R_2(\mathsf{h}, x), R(\mathsf{h}, y)$ are learned by QL-GNN, $R_1(\mathsf{h}, x) \land R_2(\mathsf{h}, y), \exists^{\geq N} y\, (R_i(y, x) \land R(\mathsf{h}, y))$ are learned by QL-GNN.*

Theorem 3.2 (proved in Appendix C) provides the logical expressivity of QL-GNN with rule formula $R(\mathsf{h}, x)$ in $\text{CML}[G, \mathsf{h}]$, which shows that querying labeling transforms $R(x, y)$ to $R(\mathsf{h}, x)$ and enable QL-GNN to learn the corresponding rule structure. To gain a concrete understanding of the rule structures learned by QL-GNN, Corollary 3.3 provides the recursive definition for these rule structures. Note that Theorem 3.2 cannot be directly applied to analyze the expressivity of QL-GNN when learning more than one rule structures. The ability of learning more than one rule structures relates to the capacity of QL-GNN, which we take as a future direction. Theorem 3.2 also reveals the maximum expressivity that QL-GNN can generalize through training, and its proof also provides some insights about the design QL-GNN with better generalization (more discussions are provided in Appendix F.1). Besides, our results in this section can be reduced to single relational-graph by restricting the relation type to a single relation type, and we give these results as corollaries in Appendix E.

### 3.2.2 EXAMPLES

We analyze several rule structures and their corresponding rule formulas in Figure 2 as illustrative examples, demonstrating the application of our theory in analyzing the rule structures that QL-GNN can learn. The real examples of these rule structures are shown in Figure 1. In Appdendix A, we have detailed analysis of rule structures discussed in the paper and present some rules from real datasets.

Chain-like rules, e.g., $C_3(x, y)$ in Figure 2, are basic rule structures investigated in many previous works (Sadeghian et al., 2019; Teru et al., 2020; Zhu et al., 2021). QL-GNN assigns constant $\mathsf{h}$ to

---

[1]The rule formula $R(\mathsf{h}, x)$ is equivalent to $\exists z R(z, x) \land P_h(z)$ where $P_h(x)$ denotes the assignment of constant $\mathsf{h}$ to $x$ and is called constant predicate in our paper.

[2]The initial representation of an entity should be unique among all entities to be regarded as constant in logic. The initial representation assigned to query entity are indeed unique in NBFNet, RED-GNN and their variants.

query entity $h$, thus triplets with relation $C_3$ can be predicted by learning the rule formula $C_3(h, x)$. $C_3(h, x)$ are formulas in $\text{CML}[G, h]$ and can be recursively defined with rules in Corollary 3.3 (proven in Corollary A.2). Therefore, our theory gives a general proof of QL-GNN's ability to learn chain-like structures.

The second type of rule structure $I_1(h, x)$ in Figure 2 is composed of a chain-like structure from query entity to tail entity along with additional entity $z_2$ connected to the chain. $I_1(h, x)$ are formulas in $\text{CML}[G, h]$ and can be defined with recursive rules in Corollary 3.3 (proven in Corollary A.3), which indicates that $I_1(h, x)$ can be learned by QL-GNN. These structures are important in KG reasoning because the entity connected to the chain can bring extra information about property of the entity it connected to (see examples of rule in Appendix A).

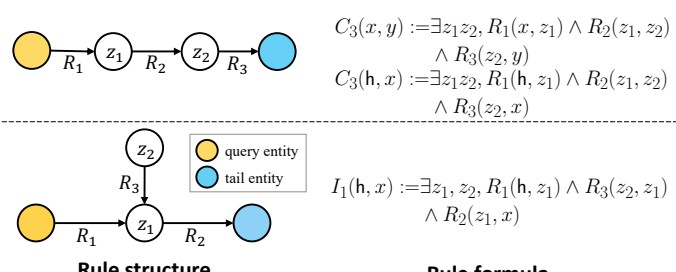

$$C_3(x, y) := \exists z_1 z_2, R_1(x, z_1) \wedge R_2(z_1, z_2) \\ \wedge R_3(z_2, y)$$
$$C_3(h, x) := \exists z_1 z_2, R_1(h, z_1) \wedge R_2(z_1, z_2) \\ \wedge R_3(z_2, x)$$

$$I_1(h, x) := \exists z_1, z_2, R_1(h, z_1) \wedge R_3(z_2, z_1) \\ \wedge R_2(z_1, x)$$

**Rule structure**  **Rule formula**

Figure 2: Example of rule structures and their corresponding rule formulas QL-GNN can learn.

## 3.3 COMPARISON WITH CLASSICAL METHODS

Classical methods such as R-GCN and CompGCN perform KG reasoning by first applying MPNN (1) to compute the entity representations $\mathbf{e}_v^{(L)}, v \in \mathcal{V}$ and then scoring the triplet $(h, R, t)$ by $s(h, R, t) = \text{Agg}(\mathbf{e}_h^{(L)}, \mathbf{e}_t^{(L)})$ with aggregation function $\text{Agg}(\cdot, \cdot)$. For simplicity, we take CompGCN as an example to analyze the expressivity of the classical methods on learning rule structures.

Since CompGCN scores a triplet using its query and tail entity representations without applying labeling trick, the rule structures learned by CompGCN should be in the form of $R(x, y)$. In CompGCN, the query and tail entities' representations encode different subgraphs. However, the joint subgraph they represent may not necessarily be connected. This suggests that the rule structures learned by CompGCN are non-structural, indicating there is no path between its query and tail entities except for relation $R$. This observation is proven with the following theorem.

**Theorem 3.4** (Logical expressivity of CompGCN). *For KG reasoning, CompGCN can learn the rule formula $R(x, y) = f_R(\{\varphi(x)\}, \{\varphi'(y)\})$ where $f_R$ is a formula involving sub-formulas from $\{\varphi(x)\}$ and $\{\varphi'(y)\}$ which are the sets of formulas in $\text{CML}[G]$.*

*Remark.* Theorem 3.4 indicates that representations of two end entities encoding two formulas respectively, and these two formulas are independent. Thus, the rule structures learned by CompGCN should be two disconnected subgraphs surrounding the query and tail entities respectively.

Similar to Theorem 3.2, CompGCN learns rule formula $R(x, y)$ by treating it as a binary classifier. In a KG, the binary classifier $R(x, y)$ should output true if the triplet $(h, R, t)$ exists; otherwise, it should output false. If CompGCN can learn the rule formula $R(x, y)$, it implies that it can estimate the binary classifier $R(x, y)$. Consequently, if the rule formula $R(x, y)$ is (not) satisfied at entity pair $(h, t)$, the score $s(h, R, t)$ is a high (low) value, indicating the existence (absence) of triplet $(h, R, t)$.

Theorem 3.4 (proven in Appendix C) shows that CompGCN can only learn rule formula $R(x, y)$ for non-structural rules. One important type of relation in this category is the similarity between two entities (experiments in Appendix D.2), like $\texttt{same\_color}(x, y)$ indicating entities with the same color. However, structural rules are more commonly observed in KG reasoning (Lavrac & Dzeroski, 1994; Sadeghian et al., 2019; Srinivasan & Ribeiro, 2020). Since Theorem 3.4 indicates CompGCN fails to learn connected rule structures that are not independent, the structural rules in Figure 2 cannot be learned by CompGCN. Such a comparison shows why QL-GNN is more efficient than classical methods, e.g., R-GCN and CompGCN, in real applications. Compared with previous work on single-relational graphs, Zhang et al. (2021) shows CompGCN cannot distinguish many non-isomorphic links, while our paper derives expressivity of CompGCN for learning rule structures.

## 4 ENTITY LABELING GNN BASED ON RULE FORMULA TRANSFORMATION

QL-GNN is proven to be able to learn the class of rule structures defined in Corollary 3.3. For rule structures outside this class, we try to learn them with a novel labeling trick based on QL-GNN. Our general idea is to transform the rule structures outside this class into the rule structures in this class by adding constants to the graph. The following proposition and corollary show how to add constants to a rule structure so that it can be described by formulas in CML and how to apply labeling trick to make it learnable for QL-GNN.

**Proposition 4.1.** *Let $R(\mathsf{h}, x)$ describe a single-connected rule structure $\mathsf{G}$ in $G$. If we assign constants $\mathsf{c}_1, \mathsf{c}_2, \cdots, \mathsf{c}_k$ to all $k$ entities with out-degree larger than one in $\mathsf{G}$, the rule structure $\mathsf{G}$ can be described with a new rule formula $R'(\mathsf{h}, x)$ in $CML[G, \mathsf{h}, \mathsf{c}_1, \mathsf{c}_2, \cdots, \mathsf{c}_k]$.*

**Corollary 4.2.** *Applying labeling trick with unique initial representations to entities assigned with constants $\mathsf{c}_1, \mathsf{c}_2, \cdots, \mathsf{c}_k$ in Proposition 4.1, the rule structure $\mathsf{G}$ can be learned by QL-GNN.*

For instance, in Figure 3, the rule structure $U$ cannot be distinguished from the rule structure $T$ by recursive definition in Corollary 3.3, thus cannot be learned by QL-GNN. In this example, Proposition 4.1 suggests assigning constant $\mathsf{c}$ to the entity colored with gray in Figure 3, then a new rule formula

$$U'(\mathsf{h}, x) := R_1(\mathsf{h}, \mathsf{c}) \wedge \big( \exists z_2, z_3, R_2(\mathsf{c}, z_2) \wedge R_4(z_2, x) \wedge R_3(\mathsf{c}, z_3) \wedge R_5(z_3, x) \big)$$

in $CML[G, \mathsf{h}, \mathsf{c}]$ (Corollary A.5) can describe the rule structure of $U$. Therefore, the rule structure of $U$ can be learned with $U'(\mathsf{h}, x)$ by QL-GNN with constant $\mathsf{c}$ and cannot be learned by classical methods and vanilla QL-GNN.

---

**Algorithm 1** Entity Labeling

**Require:** query $(h, R, ?)$, knowledge graph $G$, degree threshold $d$.
1: compute the out-degree $d_v$ of each entity $v$ in $G$;
2: **for** entity $v$ in $G$ **do**
3:     **if** $d_v > d$ **then**
4:         assign a unique representation $\mathbf{e}_v^{(0)}$ to entity $v$;
5:     **end if**
6: **end for**
7: assign initial representation $\mathbf{e}_h^{(0)}$ to the query entity $h$;
8: **Return:** initial representation of all entities.

---

Based on Corollary 4.2, we need apply labeling trick to entities other than the query entities in QL-GNN to learn the rule structures outside the scope of Corollary 3.3. The new method is called Entity-Labeling (EL) GNN shown in Algorithm 1 and is different from QL-GNN in assigning constants to all the entities with out-degree larger than $d$. We choose the degree threshold $d$ as a hyperparameter because a

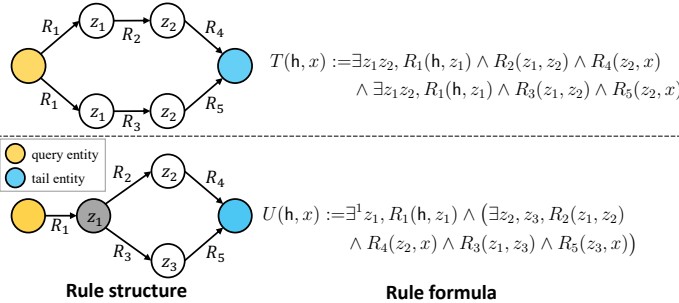

Figure 3: Two rule structures cannot be distinguished by QL-GNN.

small $d$ (such as 1) will introduce too many constants to KG, which impedes the generalization of GNN (Abboud et al., 2021) (see an explanation from logical perspective in Appendix F.2). In fact, a smaller $d$ makes GNN learn the rule formulas with many constants and results bad generalization, while a larger $d$ may not be able to transform indistinguishable rules into formulas in CML. As a result, the degree threshold $d$ should be tuned to balance the expressivity and generalization of GNN. Same as the constant $\mathsf{h}$ in QL-GNN, we add a unique initial representation $\mathbf{e}_v^{(0)}$ for entities $v$ whose out-degree $d_v > d$ in steps 3-5. For the query entity $h$, we assign it with a unique initial representation $\mathbf{e}_h^{(0)}$ in step 7. In Algorithm 1, it can be seen that the additional time of EL-GNN comes from traversing all entities in the graph. The additional time complexity is linear with respect to the number

of entities, which is negligible compared to QL-GNN. For convenience, GNN initialized with EL algorithm is denoted as EL-GNN (e.g., EL-NBFNet) in our paper.

**Discussion**    In Figure 1, we visually compare the expressivity of QL-GNN and EL-GNN. Classical methods, e.g., R-GCN and CompGCN, are not compared here because they can solely learn non-structural rules which are not commonly-seen in real applications. QL-GNN, e.g., NBFNet and RED-GNN, excels at learning rule structures described by formula $R(\mathsf{h}, x)$ in $\mathrm{CML}[G, \mathsf{h}]$. The proposed EL-GNN, encompassing QL-GNN as a special case, can learn rule structures described by formula $R(\mathsf{h}, x)$ in $\mathrm{CML}[G, \mathsf{h}, \mathsf{c}_1, \cdots, \mathsf{c}_k]$ which has a larger description scope than $\mathrm{CML}[G, \mathsf{h}]$.

## 5    RELATED WORKS

### 5.1    EXPRESSIVITY OF GRAPH NEURAL NETWORK (GNN)

GNN (Kipf & Welling, 2016; Gilmer et al., 2017) has shown good performance on a wide range of tasks involving graph-structured data, thus many existing works try to analyze the expressivity of GNNs. Most of these works analyze the expressivity of GNNs from the perspective of graph isomorphism testing. A well-known result (Xu et al., 2019) shows that the expressivity of vanilla GNN is limited to WL test and the result is extended to KG by Barcelo et al. (2022). To improve the expressivity of GNNs, most of the existing works either design GNNs motivated by high-order WL test (Morris et al., 2019; 2020; Barcelo et al., 2022) or apply special initial representations (Abboud et al., 2021; You et al., 2021; Sato et al., 2021; Zhang et al., 2021). Except for using graph isomorphism testing, Barceló et al. (2020) analyze the logical expressivity of GNNs and identify that the logical rules from graded modal logic can be learned by vanilla GNN. However, their analysis is limited to node classification on the single-relational graph. Except from the expressivity of vanilla GNN, Tena Cucala et al. (2022) propose monotonic GNN whose prediction can be explained by symbolical rules in Datalog and the expressivity of monotonic GNN is further analyzed in Cucala et al. (2023).

Regarding the expressivity of GNNs for link prediction, Srinivasan & Ribeiro (2020) demonstrate that GNNs' structural node representations alone are insufficient for accurate link prediction. To overcome this limitation, they introduce a method that incorporates Monte Carlo samples of node embeddings obtained from network embedding techniques instead of relying solely on GNNs. However, Zhang et al. (2021) discovered that by leveraging the labeling trick in GNNs, it is indeed possible to learn structural link representations for effective link prediction. This finding provides reassurance regarding the viability of GNNs for this task. Nonetheless, their analysis is confined to single-relational graphs, and their conclusions are limited to the fact that the labeling trick enables distinct representations for some non-isomorphic links, which other approaches cannot achieve. In this paper, we delve into the analysis of GNNs' logical expressivity to study their ability of learning rule structures. By doing so, we aim to gain a comprehensive understanding of the rule structures that SOTA GNNs can learn in graphs. Our analysis encompasses both single-relational graph and KGs, thus broadening the applicability of our findings.

A concurrent work by Huang et al. (2023) analyzes the expressivity of GNNs for NBFNet (a kind of QL-GNN in our paper) with conditional MPNN while our work unifies state-ot-the-art GNNs into QL-GNN and analyzes the expressivity from a different perspective focusing on the understanding of relationship between labeling trick and constants in logic.

### 5.2    KNOWLEDGE GRAPH REASONING

KG reasoning is the task to predict new facts based on the known facts in a KG $G = (\mathcal{V}, \mathcal{E}, \mathcal{R})$ where $\mathcal{V}, \mathcal{E}, \mathcal{R}$ are sets of entities, edges and relation types in the graph respectively. The facts (or edges, links) are typically expressed as triplets in the form of $(h, R, t)$, where the head entity $h$ and tail entity $t$ are related with the relation type $R$. KG reasoning can be modeled as the process of predicting the tail entity $t$ of a query in the form $(h, R, ?)$ where $h$ is called the query entity in our paper. The head prediction $(?, R, t)$ can be transformed into tail prediction $(t, R^{-1}, ?)$ with inverse relation $R^{-1}$. Thus, we focus on tail prediction in this paper.

Embedding-based methods like TransE (Bordes et al., 2013), ComplEx (Trouillon et al., 2016), RotatE (Sun et al., 2019), and QuatE (Zhang et al., 2019) have been developed for KG reasoning.

They learn embeddings for entities and relations, and predict facts by aggregating their representations. To capture local evidence within graphs, Neural LP (Yang et al., 2017) and DRUM (Sadeghian et al., 2019) learn logical rules based on predefined chain-like structures. However, apart from chain-like rules, these methods failed to learn more complex structures in KG (Hamilton et al., 2018; Ren et al., 2019). GNNs have also been used for KG reasoning, such as R-GCN (Schlichtkrull et al., 2018) and CompGCN (Vashishth et al., 2020), which aggregate entity and relation representations to calculate scores for new facts. However, these methods struggle to differentiate between the structural roles of different neighbors (Srinivasan & Ribeiro, 2020; Zhang et al., 2021). GraIL (Teru et al., 2020) addresses this by extracting enclosing subgraphs to predict new facts, while RED-GNN (Zhang & Yao, 2022) employs dynamic programming for efficient subgraph extraction and predicts new facts based on the tail entity representation. To extract relevant structures from graph, AdaProp (Zhang et al., 2023) improves RED-GNN by employing adaptive propagation to filter out irrelevant entities and retain promising targets. Motivated by the effectiveness of heuristic path-based metrics for link prediction, NBFNet (Zhu et al., 2021) proposes a neural network aligned with Bellman-Ford algorithm for KG reasoning. Zhu et al. (2022) propose A$^\star$Net to learn a priority function to select important nodes and edges at each iteration. Specifically, AdaProp and A$^\star$Net are variants of RED-GNN and NBFNet, respectively, designed to enhance their scalability. Among these methods, RED-GNN, NBFNet, AdaProp, and A$^\star$Net achieve state-of-the-art performance on KG reasoning.

## 6 EXPERIMENT

In this section, we validate our theoretical findings from Section 3 and showcase the efficacy of our proposed EL-GNN (Section 4) on synthetic and real datasets through experiments. All experiments were implemented in Python using PyTorch and executed on A100 GPUs with 80GB memory.

### 6.1 EXPERIMENTS ON SYNTHETIC DATASETS

We generate six KGs based on rule structures in Figure 2, 3, 6 to validate our theory on expressivity and verify the improved performance of EL-GNN. These rule structures are either analyzed in the previous sections, or representative for evaluating GNN's ability for learning rule structures. We evaluate R-GCN, CompGCN, RED-GNN, NBFNet, EL-RED-GNN, and EL-NBFNet (using RED-GNN/NBFNet as backbone with Algorithm 1). Our evaluation metric is prediction *Accuracy* which measures how well a rule structure is learned. We report testing accuracy of classical methods, QL-GNN, and EL-GNN on six synthetic graphs. Hyperparameters for all methods are automatically tuned with Ray (Liaw et al., 2018) based on the validation accuracy.

Table 1: Accuracy on synthetic data.

| Method | Method | $C_3$ | $C_4$ | $I_1$ | $I_2$ | $T$ | $U$ |
|---|---|---|---|---|---|---|---|
| Classical | R-GCN | 0.016 | 0.031 | 0.044 | 0.024 | 0.067 | 0.014 |
| | CompGCN | 0.016 | 0.021 | 0.053 | 0.039 | 0.067 | 0.027 |
| QL-GNN | RED-GNN | 1.0 | 1.0 | 1.0 | 1.0 | 1.0 | 0.405 |
| | NBFNet | 1.0 | 1.0 | 1.0 | 1.0 | 1.0 | 0.541 |
| EL-GNN | EL-RED-GNN | 1.0 | 1.0 | 1.0 | 1.0 | 1.0 | 0.797 |
| | EL-NBFNet | 1.0 | 1.0 | 1.0 | 1.0 | 1.0 | 0.838 |

**Dataset generation** Given a target relation, there are three steps to generate a dataset: (1) rule structure generation: generate specific rule structures according to their definition; (2) noisy triplets generation: generate noisy triplets to avoid GNN from learning naive rule structures; (3) missing triplets completion: generate missing triplets based on the target rule structure because the noisy triplets generation step could add triplets satisfying the target rule structure. We use triplets generated from rule structure and noisy triplets generation steps as known triplets in graph. Triplets with the target relation are separated into training, validation, and testing sets. Our experimental setting differs slightly from previous works as all GNNs in the experiments only perform message passing on the known triplets in the graph. This setup is reasonable and allows for evaluating the performance of GNNs in learning rule structures because the presence of a triplet can be determined based on the known triplets in the graph, following the rule structure generation process.

**Results**  Table 1 presents the testing accuracy of classical GNN methods, QL-GNN, and EL-GNN on six synthetic datasets (denoted as $C_3, C_4, I_1, I_2, T$, and $U$) generated from their corresponding rule structures. The experimental results support our theory. CompGCN performs poorly on all six datasets, as it fails to learn the underlying rule structures discussed in examples of Section 3 (refer to Section D.2 for experiments of CompGCN). QL-GNN achieves perfect predictions (100% accuracy) for triplets with relations $C_l, I_i$, and $T$, successfully learning the corresponding rule formulas from $\text{CML}[G, \mathsf{h}]$. EL-GNN demonstrates improved expressivity, as evidenced by its performance on dataset $U$, aligning with the analysis in Section 4. Furthermore, EL-GNN effectively learns rule formulas $C(\mathsf{h}, x)$ and $I(\mathsf{h}, x)$, validating its expressivity.

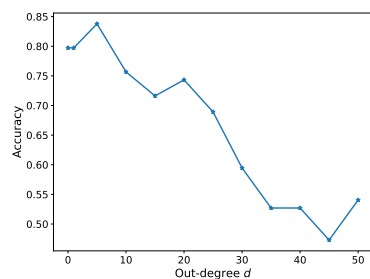

Furthermore, we demonstrate the impact of the degree threshold $d$ on EL-GNN with dataset $U$. The testing accuracy in Figure 4 reveals that an excessively small or large out-degree $d$ hinders the performance of EL-GNN. Therefore, it is important to empirically fine-tune the hyperparameter $d$. To test the robustness of QL-GNN and EL-GNN in learning rules with incomplete structures, we randomly remove triplets in the training set to evaluate the accuracy of learning rule structures. The results can be found in Appendix D.4.

Figure 4: Accuracy versus out-degree $d$ of EL-GNN on the dataset with relation $U$.

## 6.2  EXPERIMENTS ON REAL DATASETS

In this section, we follow the standard setup as Zhu et al. (2021) to test EL-GNN's effectiveness on four real datasets: Family (Kok & Domingos, 2007), Kinship (Hinton et al., 1986), UMLS (Kok & Domingos, 2007), WN18RR (Dettmers et al., 2017), and FB15k-237 (Toutanova & Chen, 2015). For a fair comparison, we evaluate EL-NBFNet and EL-RED-GNN (applying EL to NBFNet and RED-GNN) using the same hyperparameters as NBFNet and RED-GNN and handcrafted $d$. We compare it with embedding-based methods (RotatE, QuatE), rule-based methods (Neural LP, DRUM), and GNN-based methods (CompGCN, NBFNet, RED-GNN). To evaluate performance, we provide testing accuracy and standard deviation obtained from three repetitions for thorough evaluation.

In Table 2, we present our experimental findings. The results first show that NBFNet and RED-GNN (QL-GNN) outperform CompGCN. Furthermore, the proposed EL algorithm improves the accuracy of RED-GNN and NBFNet on real datasets. However, the degree of improvement varies across datasets due to the number and variations of rule types, and the quality of missing triplets in training sets. More experimental results, e.g., time cost and more performance metrics, are in Appendix D.5.

Table 2: Accuracy and standard deviation on real datasets. The best (and comparable best) results are in "**bold**", the second (and comparable second) best are underlined.

| Method Class | Methods | Family | Kinship | UMLS | WN18RR | FB15k-237 |
|---|---|---|---|---|---|---|
| Embedding-based | RotatE | 0.865±0.004 | 0.704±0.002 | 0.860±0.003 | 0.427±0.003 | 0.240±0.001 |
| | QuatE | 0.897±0.001 | 0.311±0.003 | 0.907±0.002 | 0.441±0.002 | 0.255±0.004 |
| Rule-based | Neural LP | 0.872±0.002 | 0.481±0.006 | 0.630±0.001 | 0.369±0.003 | 0.190±0.002 |
| | DRUM | 0.880±0.003 | 0.459±0.005 | 0.676±0.004 | 0.424±0.002 | 0.252±0.003 |
| GNN-based | CompGCN | 0.883±0.001 | 0.751±0.003 | 0.867±0.002 | 0.443±0.001 | 0.265±0.001 |
| | RED-GNN | **0.988±0.002** | 0.820±0.003 | 0.946±0.001 | 0.502±0.001 | 0.284±0.002 |
| | NBFNet | 0.977±0.001 | 0.819±0.002 | 0.946±0.002 | 0.496±0.002 | 0.320±0.001 |
| | EL-RED-GNN | **0.990±0.002** | **0.839±0.001** | **0.952±0.003** | **0.504±0.001** | 0.322±0.002 |
| | EL-NBFNet | **0.985±0.001** | **0.842±0.003** | **0.953±0.002** | 0.501±0.003 | **0.332±0.001** |

## 7  CONCLUSION

In this paper, we analyze the expressivity of the state-of-the-art GNNs for learning rules in KG reasoning, explaining their superior performance over classical methods. Our analysis sheds light on the rule structures that GNNs can learn. Additionally, our theory motivates an effective labeling method to improve GNN's expressivity. Moving forward, we will extend our analysis to GNNs with general labeling trick and try to extract explainable rule structures from trained GNN. Limitations and impacts are discussed in Appendix G.

ACKNOWLEDGMENTS

Q. Yao was in part supported by National Key Research and Development Program of China under Grant 2023YFB2903904 and NSFC (No. 92270106).

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

## A    RULE ANALYSIS

We first give a simple proof for Proposition 3.1.

*proof of Proposition 3.1.* Since $R(\mathsf{h}, x)$ is equivalent to $\exists z R(z, x) \wedge P_h(z)$, where $P_h(z)$ is the constant predicate only satisfied at entity $h$. Because $R(z, x)$ can describe the rule structure of $(h, R, ?)$, $\exists z R(z, x) \wedge P_h(z)$ can describe the rule structure of $(h, R, ?)$ as well. $\square$

We use the notation $G, v \models P_i$ ($G, v \nvDash P_i$) to represent that the unary predicate $P_i(x)$ is (not) satisfied at entity $v$.

**Definition A.1** (Definition of graded modal logic). A formula in graded modal logic of KG $G$ is recursively defined as follows:

1. If $\varphi(x) = \top$, $G, v \models \varphi$ if $v$ is an entity in KG;

2. If $\varphi(x) = P_c(x)$, $G, v \models \varphi$ if and only if $v$ has the property $P_c$ or can be uniquely identified by constant c;

3. If $\varphi(x) = \varphi_1(x) \wedge \varphi_2(x)$, $G, v \models \varphi$ if and only if $G, v \models \varphi_1$ and $G, v \models \varphi_2$;

4. If $\varphi(x) = \neg\phi(x)$, $G, v \models \varphi$ if and only if $G, v \nvDash \phi$;

5. If $\varphi(x) = \exists^{\geq N} y, R_j(y, x) \wedge \phi(y)$, $G, v \models \varphi$ if and only if the set of entities $\{u | u \in \mathcal{N}_{R_j}(v) \text{ and } G, u \models \phi\}$ has cardinality at least $N$.

**Corollary A.2.** $C_3(\mathsf{h}, x)$ *are formulas in CML$[G, \mathsf{h}]$.*

*Proof.* $C_3(\mathsf{h}, x)$ is a formula in CML$[G, \mathsf{h}]$ as it can be recursively defined as follows

$$\varphi_1(x) = P_h(x),$$
$$\varphi_2(x) = \exists y, R_1(y, x) \wedge \varphi_1(y),$$
$$\varphi_3(x) = \exists y, R_2(y, x) \wedge \varphi_2(y),$$
$$C_3(\mathsf{h}, x) = \exists y, R_3(y, x) \wedge \varphi_3(y).$$

$\square$

**Corollary A.3.** $I_1(\mathsf{h}, x)$ *is a formula in CML$[G, \mathsf{h}]$.*

*Proof.* $I_1(\mathsf{h}, x)$ is a formula in CML$[G, \mathsf{h}]$ as it can be recursively defined as follows

$$\varphi_1(x) = P_h(x),$$
$$\varphi_2(x) = \exists y, R_1(y, x) \wedge \varphi_1(y),$$
$$\varphi_s(x) = \exists y, R_3(y, x) \wedge \top,$$
$$\varphi_3(x) = \varphi_s(x) \wedge \varphi_2(x),$$
$$I_1(\mathsf{h}, x) = \exists y, R_2(y, x) \wedge \varphi_3(y).$$

$\square$

**Corollary A.4.** $T(\mathsf{h}, x)$ *is a formula in CML$[G, \mathsf{h}]$.*

*Proof.* By Corollary A.2, $C_3'(\mathsf{h}, x) := \exists z_1 z_2, R_1(\mathsf{h}, z_1) \wedge R_2(z_1, z_2) \wedge R_4(z_2, x)$ and $C_3^\star(\mathsf{h}, x) := \exists z_1 z_2, R_1(\mathsf{h}, z_1) \wedge R_3(z_1, z_2) \wedge R_5(z_2, x)$ are formulas in CML$[G, \mathsf{h}]$. Thus $T(\mathsf{h}, x) = C_3'(\mathsf{h}, x) \wedge C_3^\star(\mathsf{h}, x)$ is a formula in CML$[G, \mathsf{h}]$. $\square$

**Corollary A.5.** $U'(\mathsf{h}, x)$ *is a formula in CML$[G, \mathsf{h}, \mathsf{c}]$.*

*Proof.* $U'(\mathsf{h}, x)$ is a formula in $\mathrm{CML}[G, \mathsf{h}, \mathsf{c}]$ as it can be recursively defined as follows

$$\varphi_1(x) = P_h(x), \varphi_c(x) = P_c(x),$$
$$\varphi_2(x) = \exists y, R_1(y, x) \wedge \varphi_1(y),$$
$$\varphi_3(x) = \varphi_2(x) \wedge \varphi_c(x),$$
$$\varphi'_4(x) = \exists y, R_2(y, x) \wedge \varphi_3(y),$$
$$\varphi'_5(x) = \exists y, R_4(y, x) \wedge \varphi'_4(y),$$
$$\varphi''_4(x) = \exists y, R_3(y, x) \wedge \varphi_3(y),$$
$$\varphi''_5(x) = \exists y, R_5(y, x) \wedge \varphi''_4(y),$$
$$U'(\mathsf{h}, x) = \varphi'_5(x) \wedge \varphi''_5(x)$$

where the constant $\mathsf{c}$ ensures that there is only one entity satisfied for unary predicate $\varphi_3(x)$. $\square$

**Example of rules** We can find some relations in reality corresponding to rules in Figure 2. Here are two examples of $C_3$ and $I_1$:

- Relation `nationality` ($C_3$): Einstein $\rightarrow_{\text{born\_in}}$ Ulm $\rightarrow_{\text{hometown\_of}}$ Born $\rightarrow_{nationality}$ Germany;
- Relation `father` ($I_1$): A $\rightarrow_{\text{spouse}}$ B $\rightarrow_{\text{parent}}$ C and D $\rightarrow_{\text{sisterhood}}$ B.

**Rule structures in real datasets** To show that the expressivity is meaningful in our paper, we select three rule structures from Family and FB15k-237 in Figure 5 to show the existence of rule structures in real datasets. With the definition of CML, the rule structure in Figure 5(a) is not a formula in CML and rule structures in Figure 5(b) and 5(c) are formulas in CML. The real rules shows that rules defined by CML is common in real-world datasets and the rules beyond CML also exist, which highlights the importance of our work.

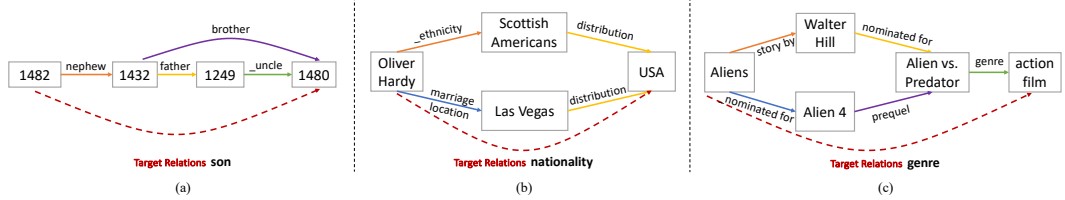

Figure 5: Some rule structures in real datasets. The rule structure (a) is from Family dataset and is not a rule formula in $\mathrm{CML}[G, \mathsf{h}]$, which cannot not be learned by QL-GNN. The rule structures (b) and (c) are from FB15k-237 dataset and are rule formulas in $\mathrm{CML}[G, \mathsf{h}]$, which can be learned by QL-GNN.

**Summary** Here we give Table 3 to illustrate the correspondence between GNNs for KG reasoning, rule structures, and theories presented in our paper.

Table 3: Whether GNNs investigated in our paper can learn the rule formulas in Figure 2 and 3 and the exemplar methods of these GNNs. ✓(✗) mean the corresponding GNN can(not) lean the rule formula.

| Rule formula | $C_3(\mathsf{h}, x)$ | $I_1(\mathsf{h}, x)$ | $T(\mathsf{h}, x)$ | $U(\mathsf{h}, x)$ | Theoretical result | Exemplar Methods |
|---|---|---|---|---|---|---|
| Classical | ✗ | ✗ | ✗ | ✗ | Theorem 3.4 | R-GCN, CompGCN |
| QL-GNN | ✓ | ✓ | ✓ | ✗ | Theorem 3.2 | NBFNet, RED-GNN |
| EL-GNN | ✓ | ✓ | ✓ | ✓ | Proposition 4.1 | EL-NBFNet/RED-GNN |

## B RELATION BETWEEN QL-GNN AND NBFNET/RED-GNN

In this part, we show that NBFNet and RED-GNN are special cases of QL-GNN in Table 4 and 5 respectively.

Table 4: NBFNet is a special case of QL-GNN.

| | NBFNet |
|---|---|
| Query representation | Relation embedding |
| Non-query representation | 0 |
| MPNN | $\text{AGGREGATE}\left(\left\{\text{MESSAGE}\left(\boldsymbol{h}_x^{(t-1)}, \boldsymbol{w}_q(x, r, v)\right)\Big\|(x, r, v) \in \mathcal{E}(v)\right\} \cup \left\{\boldsymbol{h}_v^{(0)}\right\}\right)$ |
| Triplet score | Feed-forward network |

Table 5: RED-GNN is a special case of QL-GNN.

| | RED-GNN |
|---|---|
| Query representation | 0 |
| Non-query representation | NULL |
| MPNN | $\delta\left(\sum_{\{e_s, r\}:(e_s, r, e) \in \mathcal{E}_{e_q}^\ell} \varphi\left(\boldsymbol{h}_{e_q, e_s}^{\ell-1}, \boldsymbol{h}_r^\ell\right)\right)$ |
| Triplet score | Linear transformation |

## C  PROOF

We use the notation $G, (h, t) \models R_j$ ($G, (h, t) \nvDash R_j$) to denote $R_j(x, y)$ is (not) satisfied at $h, t$.

### C.1  BASE THEOREM: WHAT KIND OF LOGICAL FORMULAS CAN MPNN BACKBONE FOR KG LEARN?

In this section, we analyze the expressivity of MPNN backbone (1) for learning logical formulas in KG. This section is the extension of Barceló et al. (2020) to KG.

In a KG $G = (\mathcal{V}, \mathcal{E}, \mathcal{R})$, MPNN with $L$ layers is a type of neural network that applies graph $G$ and initial entity representation $\mathbf{e}_v^{(0)}$ to learn the representations $\mathbf{e}_v^{(L)}, v \in \mathcal{V}$. MPNN employs message-passing mechanisms (Gilmer et al., 2017) to propagate information between entities in graph. The $k$-th layer of MPNN updates the entity representation via the following message-passing formula

$$\mathbf{e}_v^{(k)} = \delta\left(\mathbf{e}_v^{(k-1)}, \phi\left(\{\{\psi(\mathbf{e}_u^{(k-1)}, R)|u \in \mathcal{N}_R(v), R \in \mathcal{R}\}\}\right)\right),$$

where $\delta$ and $\phi$ are combination and aggregation functions respectively, $\psi$ is the message function encoding the relation $R$ and entity $u$ neighboring to $v$, $\{\{\cdots\}\}$ is a multiset, and $\mathcal{N}_R(v)$ is the neighboring entity set $\{u|(u, R, v) \in \mathcal{E}\}$.

To understand how MPNN can learn logical formulas, we regard logical formula $\varphi(x)$ as a binary classifier indicating whether $\varphi(x)$ is satisfied at entity $x$. Then, we commence with the following definition.

**Definition C.1.** A MPNN captures a logical formula $\varphi(x)$ if and only if given any graph $G$, the MPNN representation can be mapped to a binary value, where `True` indicates that $\varphi(x)$ satisfies on entity $x$, while `False` does not satisfy.

According to the above definition, MPNN can learn logical formula in KG by encoding whether these logical formulas is satisfied in the representation of the corresponding entity. For example, if MPNN can learn a logical formula $\varphi(x)$, it implies that $\mathbf{e}_v^{(L)}$ can be mapped to a binary value `True`/`False` by a function indicating whether $\varphi(x)$ is satisfied at entity $v$. Previous work (Barceló et al., 2020) has proven that vanilla GNN for single-relational graph can learn the logical formulas from graded modal logic (De Rijke, 2000; Otto, 2019) (a.k.a., Counting extension of Modal Logic, CML). In this section, we will present a similar theory of MPNN for KG.

The insight of MPNN's ability to learn formulas in CML lies in the alignment between certain CML formulas and the message-passing mechanism, which also holds for KG. Specifically, $\exists^{\geq N} y\left(R_j(y, x) \wedge \varphi(y)\right)$ is the formula aligned with MPNN's message-passing mechanism and allows to check the property of neighbor $y$ of entity variable $x$. We use notation $\text{CML}[G]$ to denote

CML of a graph $G$. Then, we give the following theorem to find out the kind of logical formula MPNN (1) can learn in KG.

**Theorem C.2.** *In a KG $G$, a logical formula $\varphi(x)$ is learned by MPNN (1) from its representations if and only if $\varphi(x)$ is a formula in CML$[G]$.*

Our theorem can be viewed as an extension of Theorem 4.2 in Barceló et al. (2020) to KG and is the elementary tool for analyzing the expressivity of GNNs for KG reasoning. The proof of Theorem C.2 is in Appendix C and employs novel techniques that specifically account for relation types. Our theorem shows that CML of KG is the tightest subclass of logic that MPNN can learn. Similarly, our theorem is about the ability to implicitly learn logical formulas by MPNN rather than explicitly extracting them.

## C.2 PROOF OF THEOREM C.2

The backward direction of Theorem C.2 is proven by constructing a MPNN that can learn any formula $\varphi(x)$ in CML. The forward direction relies on the results from recent theoretical results in Otto (2019). Our theorem can be seen as an extension of Theorem 4.2 in Barceló et al. (2020) to KG.

We first prove the backward direction of Theorem C.2.

**Lemma C.3.** *Each formula $\varphi(x)$ in CML can be learned by MPNN (1) from its entity representations.*

*Proof.* Let $\varphi(x)$ be a formula in CML. We decompose $\varphi$ into a series of sub-formulas $\text{sub}[\varphi] = (\varphi_1, \varphi_2, \cdots, \varphi_L)$ where $\varphi_k$ is a sub-formula of $\varphi_\ell$ if $k \leq \ell$ and $\varphi = \varphi_L$. Assume the MPNN representation $\mathbf{e}_v^{(i)} \in \mathbb{R}^L, v \in \mathcal{V}, i = 1 \cdots L$. In this proof, the theoretical analysis will based on the following simple choice of (1)

$$\mathbf{e}_v^{(i)} = \sigma \left( \mathbf{e}_v^{(i-1)} \mathbf{C} + \sum_{j=1}^{r} \sum_{u \in \mathcal{N}_{R_j}(v)} \mathbf{e}_u^{(i-1)} \mathbf{A}_{R_j} + \mathbf{b} \right) \tag{2}$$

with $\sigma = \min(\max(0, x), 1)$, $\mathbf{A}_{R_j}, \mathbf{C} \in \mathbb{R}^{L \times L}$ and $\mathbf{b} \in \mathbb{R}^L$. The entries of the $\ell$-th columns of $\mathbf{A}_{R_j}, \mathbf{C}$, and $\mathbf{b}$ depend on the sub-formulas of $\varphi$ as follows:

- **Case 0.** if $\varphi_\ell(x) = P_\ell(x)$ where $P_\ell$ is a unary predicate, $\mathbf{C}_{\ell\ell} = 1$;

- **Case 1.** if $\varphi_\ell(x) = \varphi_j(x) \wedge \varphi_k(x)$, $\mathbf{C}_{j\ell} = \mathbf{C}_{k\ell} = 1$ and $\mathbf{b}_\ell = -1$;

- **Case 2.** if $\varphi_\ell = \neg\varphi_k(x)$, $\mathbf{C}_{k\ell} = -1$ and $\mathbf{b}_\ell = 1$;

- **Case 3.** if $\varphi_\ell(x) = \exists^{\geq N} y \left( R_j(y, x) \wedge \varphi_k(y) \right)$, $\left( \mathbf{A}_{R_j} \right)_{k\ell} = 1$ and $\mathbf{b}_\ell = -N + 1$.

with all the other values set to 0.

Before the proof, for every entity $v \in \mathcal{V}$, the initial representation $\mathbf{e}_v^{(0)} = (t_1, t_2, \cdots, t_n)$ has $t_\ell = 1$ if the sub-formula $\varphi_\ell = P_\ell(x)$ is satisfied at $v$, and $t_\ell = 0$ otherwise.

Let $G = (\mathcal{V}, \mathcal{E}, \mathcal{R})$ be a KG. We next prove that for every $\varphi_\ell \in \text{sub}[\varphi]$ and every entity $v \in \mathcal{V}$ it holds that

$$\left( \mathbf{e}_v^{(i)} \right)_\ell = 1 \quad \text{if} \quad G, v \models \varphi_\ell, \quad \text{and} \quad \left( \mathbf{e}_v^{(i)} \right)_\ell = 0 \quad \text{otherwise,}$$

for every $\ell \leq i \leq L$.

Now, we prove this by induction of the number of formulas in $\varphi$.

**Base case:** One sub-formula in $\varphi$. In this case, the formula is an atomic predicate $\varphi = \varphi_\ell(x) = P_\ell(x)$. Because $\mathbf{C}_{\ell\ell} = 1$ and $(\mathbf{e}_v^{(0)})_\ell = 1, (\mathbf{e}_v^{(0)})_i = 0, i \neq \ell$, we have $(\mathbf{e}_v^{(1)})_\ell = 1$ if $G, v \models \varphi_\ell$ and $(\mathbf{e}_v^{(1)})_\ell = 0$ otherwise. For $i \geq 1$, $\mathbf{e}_v^{(i)}$ satisfies the same property.

**Induction Hypothesis:** $k$ sub-formulas in $\varphi$ with $k < \ell$. Assume $\left( \mathbf{e}_v^{(i)} \right)_k = 1$ if $G, v \models \varphi_k$ and $\left( \mathbf{e}_v^{(i)} \right)_k = 0$ otherwise for $k \leq i \leq L$.

**Proof:** $\ell$ sub-formulas in $\varphi$. Let $i \geq \ell$. Case 1-3 should be considered.

Case 1. Let $\varphi_\ell(x) = \varphi_j(x) \wedge \varphi_k(x)$. Then $\mathbf{C}_{j\ell} = \mathbf{C}_{k\ell} = 1$ and $\mathbf{b}_\ell = -1$. Then we have

$$(\mathbf{e}_v^{(i)})_\ell = \sigma\left((\mathbf{e}_v^{(i-1)})_j + (\mathbf{e}_v^{(i-1)})_k - 1\right).$$

By the induction hypothesis, $(\mathbf{e}_v^{(i-1)})_j = 1$ if only if $G, v \models \varphi_j$ and $(\mathbf{e}_v^{(i-1)})_j = 0$ otherwise. Similarly, $(\mathbf{e}_v^{(i-1)})_k = 1$ if and only if $G, v \models \varphi_k$ and $(\mathbf{e}_v^{(i-1)})_k = 0$ otherwise. Then we have $(\mathbf{e}_v^{(i)})_\ell = 1$ if and only if $(\mathbf{e}_v^{(i-1)})_j + (\mathbf{e}_v^{(i-1)})_k - 1 \geq 1$, which means $(\mathbf{e}_v^{(i-1)})_j = 1$ and $(\mathbf{e}_v^{(i-1)})_k = 1$. Then $(\mathbf{e}_v^{(i)})_\ell = 1$ if and only if $G, v \models \varphi_j$ and $G, v \models \varphi_k$, i.e., $G, v \models \varphi_\ell$, and $(\mathbf{e}_v^{(i)})_\ell = 0$ otherwise.

Case 2. Let $\varphi_\ell(x) = \neg\varphi_k(x)$. Because of $\mathbf{C}_{k\ell} = -1$ and $\mathbf{b}_\ell = 1$, we have

$$(\mathbf{e}_v^{(i)})_\ell = \sigma\left(-(\mathbf{e}_v^{(i-1)})_k + 1\right).$$

By the induction hypothesis, $(\mathbf{e}_v^{(i-1)})_k = 1$ if and only if $G, v \models \varphi_k$ and $(\mathbf{e}_v^{(i-1)})_k = 0$ otherwise. Then we have $(\mathbf{e}_v^{(i)})_\ell = 1$ if and only if $-(\mathbf{e}_v^{(i-1)})_k + 1 \geq 1$, which means $(\mathbf{e}_v^{(i-1)})_k = 0$. Because $(\mathbf{e}_v^{(i-1)})_k = 0$ if and only if $G, v \nvDash \varphi_k$, we have $(\mathbf{e}_v^{(i)})_\ell = 1$ if and only if $G, v \nvDash \varphi_k$, i.e., $G, v \models \varphi_\ell$, and $(\mathbf{e}_v^{(i)})_\ell = 0$ otherwise.

Case 3. Let $\varphi_\ell(x) = \exists^{\geq N} y\, (R_j(y, x) \wedge \varphi_k(y))$. Because of $(\mathbf{A}_{R_j})_{k\ell} = 1$ and $\mathbf{b}_\ell = -N + 1$, we have

$$(\mathbf{e}_v^{(i)})_\ell = \sigma\left(\sum_{u \in \mathcal{N}_{R_j}(v)} (\mathbf{e}_u^{(i-1)})_k - N + 1\right).$$

By the induction hypothesis, $(\mathbf{e}_u^{(i-1)})_k = 1$ if and only if $G, u \models \varphi_k$ and $(\mathbf{e}_u^{(i-1)})_k = 0$ otherwise. Let $m = |\{u | u \in \mathcal{N}_{R_j}(v) \text{ and } G, u \models \varphi_k\}|$. Then we have $(\mathbf{e}_v^{(i)})_\ell = 1$ if and only if $\sum_{u \in \mathcal{N}_{R_j}(v)} (\mathbf{e}_u^{(i-1)})_k - N + 1 \geq 1$, which means $m \geq N$. Because $G, u \models \varphi_k$, $u$ is connected to $v$ with relation $R_j$, and $m \geq N$, we have $(\mathbf{e}_v^{(i)})_\ell = 1$ if and only if $G, v \models \varphi_\ell$ and $(\mathbf{e}_v^{(i)})_\ell = 0$ otherwise.

To learn a logical formula $\varphi(x)$, we only apply a linear classifier to $\mathbf{e}_v^{(L)}, v \in \mathcal{V}$ to extract the component of $\mathbf{e}_v^{(L)}$ corresponding to $\varphi$. If $G, v \models \varphi$, the value of the corresponding extracted component is 1.

□

Next, we prove the forward direction of Theorem C.2.

**Theorem C.4.** *A formula $\varphi(x)$ is learned by MPNN (1) if it can be expressed as a formula in CML.*

To prove Theorem C.4, we introduce Definition C.5, Lemma C.6, Theorem C.7, and Lemma C.8.

**Definition C.5** (Unraveling tree). Let $G$ be a KG, $v$ be entity in $G$, and $L \in \mathbb{N}$. The unravelling of $v$ in $G$ at depth $L$, denoted by $\mathrm{Unr}_G^L(v)$, is a tree composed of

- a node $(v, R_1, u_1, \cdots, R_i, u_i)$ for each path $(v, R_1, u_1, \cdots, R_i, u_i)$ in $G$ with $i \leq L$,

- an edge $R_i$ between $(v, R_1, u_1, \cdots, R_{i-1}, u_{i-1})$ and $(v, R_1, u_1, \cdots, R_i, u_i)$ when $(u_i, R_i, u_{i-1})$ is a triplet in $G$ (assume $u_0$ is $v$), and

- each node $(v, R_1, u_1, \cdots, R_i, u_i)$ has the same properties as $u_i$ in $G$.

**Lemma C.6.** *Let $G$ and $G'$ be two KGs, $v$ and $v'$ be two entities in $G$ and $G'$ respectively. Then for every $L \in \mathbb{N}$, the RWL test (Barcelo et al., 2022) assigns the same color/hash to $v$ and $v'$ at round $L$ if and only if there is an isomorphism between $\mathrm{Unr}_G^L(v)$ and $\mathrm{Unr}_{G'}^L(v')$ sending $v$ to $v'$.*

*Proof.* **Base Case:** When $L = 1$, the result is obvious.

**Induction Hypothesis:** Relational WL (RWL) test assigns the same color to $v$ and $v'$ at round $L-1$ if and only if there is an isomorphism between $\text{Unr}_G^{L-1}(v)$ and $\text{Unr}_{G'}^{L-1}(v')$ sending $v$ to $v'$.

**Proof:** In the $L$-th round,

$\bullet$ Prove "same color $\Rightarrow$ isomorphism".

$$c^L(v) = \text{hash}(c^{L-1}(v), \{\{(c^{L-1}(u), R_i)|u \in \mathcal{N}_{R_i}(v), i = 1, \cdots, r\}\}),$$
$$c^L(v') = \text{hash}(c^{L-1}(v'), \{\{(c^{L-1}(u'), R_i)|u \in \mathcal{N}_{R_i}(v'), i = 1, \cdots, r\}\}).$$

Because $c^L(v) = c^L(v')$, we have $c^{L-1}(v) = c^{L-1}(v')$, and there exists an entity pair $(u, u'), u \in \mathcal{N}_{R_i}(v), u' \in \mathcal{N}_{R_i}(v')$ that

$$(c^{L-1}(u), R_i) = (c^{L-1}(u'), R_i).$$

Then we have $c^{L-1}(u) = c^{L-1}(u')$. According to induction hypothesis, we have $\text{Unr}_G^{L-1}(u) \cong \text{Unr}_{G'}^{L-1}(u')$. Also, because the edge connecting entity pair $(v, u)$ and $(v', u')$ is $R_i$, so there is an isomorphism between $\text{Unr}_G^L(v)$ and $\text{Unr}_{G'}^L(v')$ sending $v$ to $v'$.

$\bullet$ Prove "isomorphism $\Rightarrow$ same color".

Because there exists an isomorphism $\pi$ between $\text{Unr}_G^L(v)$ and $\text{Unr}_{G'}^L(v')$ sending $v$ to $v'$, assume $\pi$ is an bijective between the neighbors of $v$ and $v'$, e.g, $u \in \mathcal{N}_{R_i}(v), u' \in \mathcal{N}_{R_i}(v')$ and $u'_i = \pi(u_i)$, the relation between entity pair $(u, v)$ and $(u', v')$ is $R_i$.

Next we prove $c^{L-1}(u) = c^{L-1}(u')$. Because $\text{Unr}_G^L(v)$ and $\text{Unr}_G^L(v')$ are isomorphism, and $\pi$ maps $u \in \mathcal{N}_{R_i}(v)$ to $u' \in \mathcal{N}_{R_i}(v')$, for the left tree with $L-1$ depth, i.e., $\text{Unr}_G^{L-1}(u)$ and $\text{Unr}_{G'}^{L-1}(u')$, $\pi$ can be the isomorphism mapping between $\text{Unr}_G^{L-1}(u)$ and $\text{Unr}_{G'}^{L-1}(u')$. According to induction hypothesis, we have $c^{L-1}(u) = c^{L-1}(u')$. Because $\text{Unr}_G^L(v) \cong \text{Unr}_{G'}^L(v')$, we also have $\text{Unr}_G^{L-1}(u) \cong \text{Unr}_{G'}^{L-1}(u')$ which means $c^{L-1}(u) = c^{L-1}(u')$. After running RWL test, we have $c^L(v) = c^L(v')$. $\qquad\square$

**Theorem C.7.** *Let $\varphi(x)$ be a unary formula in the formal description of graph $G$ in Section 3.1. If $\varphi(x)$ is not equivalent to a formula in CML, there exist two KGs $G$ and $G'$ and two entities $v$ in $G$ and $v'$ in $G'$ such that $Unr_G^L(v) \cong Unr_{G'}^L(v')$ for every $L \in \mathbb{N}$ and such that $G, v \models \varphi$ but $G', v' \nvDash \varphi$.*

*Proof.* The theorem follows directly from Theorem 2.2 in Otto (2019). Because $G, v \sim_\# G', v'$ and $\text{Unr}_G^L(v) \cong \text{Unr}_{G'}^L(v')$ are equivalent with the definition of counting bisimulation (i.e., notation $\sim_\#$). $\qquad\square$

**Lemma C.8.** *If a formula $\varphi(x)$ is not equivalent to any formula in CML, there is no MPNN (1) that can learn $\varphi(x)$.*

*Proof.* Assume for a contradiction that there exists a MPNN that can learn $\varphi(x)$. Since $\varphi(x)$ is not equivalent to any formula in CML, with Theorem C.7, there exists two KGs $G$ and $G'$ and two entities $v$ in $G$ and $v'$ in $G'$ such that $\text{Unr}_G^L(v) \cong \text{Unr}_{G'}^L(v')$ for every $L \in \mathbb{N}$ and such that $G, v \models \varphi$ and $G', v' \nvDash \varphi$. By Lemma C.6, because $\text{Unr}_G^L(v) \cong \text{Unr}_{G'}^L(v')$ for every $L \in \mathbb{N}$, we have $\mathbf{e}_v^{(L)} = \mathbf{e}_{v'}^{(L)}$. But this contradicts the assumption that MPNN is supposed to learn $\varphi(x)$. $\qquad\square$

*Proof of Theorem C.4.* Theorem can be obtained directly from Lemma C.8. $\qquad\square$

*Proof of Theorem C.2.* Theorem can be obtained directly by combining Lemma C.3 and Theorem C.4. $\qquad\square$

The following two remarks intuitively explain why MPNN can learn formulas in CML.

*Remark* C.9. Theorem C.2 applies to both CML$[G]$ and CML$[G, \mathsf{c}_1, \mathsf{c}_2, \cdots, \mathsf{c}_k]$. The atomic unary predicate $P_i(x)$ in CML of graph $G$ is learned by the initial representations $\mathbf{e}_v^{(0)}, v \in \mathcal{V}$, which can be achieved by assigning special vectors to $\mathbf{e}_v^{(0)}, v \in \mathcal{V}$. In particular, the constant predicate $P_c(x)$ in CML$[G, \mathsf{c}]$ is learned by assigning a unique vector (e.g., one-hot vector for different entities) as the initial representation of the entity with unique identifier $\mathsf{c}$. The other sub-formulas $\neg\varphi(x), \varphi_1(x) \wedge \varphi_2(x)$ in Definition A.1 can be learned by continuous logical operations (Arakelyan et al., 2021) which are independent of message-passing mechanisms.

*Remark* C.10. Assume the $(i-1)$-th layer representations $\mathbf{e}_v^{(i-1)}, v \in \mathcal{V}$ can learn the formula $\varphi(x)$ in CML, the $i$-th layer representations $\mathbf{e}_v^{(i)}, v \in \mathcal{V}$ of MPNN can learn $\exists^{\geq N} y, R_j(y, x) \wedge \varphi(y)$ with specific aggregation function in (1) because $\mathbf{e}_v^{(i)}, v \in \mathcal{V}$ can aggregate the logical formulas in the one-hop neighbor representation $\mathbf{e}_v^{(i-1)}, v \in \mathcal{V}$ (i.e., $\varphi(x)$) with message-passing mechanisms.

The following remark clarifies the scope of Theorem C.2 and 3.2.

*Remark* C.11. The positive results for our theorem (e.g., a MPNN variant can learn a logical formula) hold for MPNNs powerful than the MPNN we construct in (2), while our negative results (e.g., a MPNN variant cannot learn a logical formula) hold for any general MPNNs (1). Hence, the backward direction remains valid irrespective of the aggregate and combine operators under consideration. This limitation is inherent to the MPNN architecture represented by (1) and not specific to the chosen representation update functions. On the other hand, the forward direction holds for MPNNs that are more powerful than (2).

## C.3 Proof of Theorem 3.2

**Definition C.12.** QL-GNN learns a rule formula $R(\mathsf{h}, x)$ if and only if given any graph $G$, the QL-GNN's score of a new triplet $(h, R, t)$ can be mapped to a binary value, where `True` indicates that $R(\mathsf{h}, x)$ satisfies on entity $t$, while `False` does not satisfy.

*Proof.* We set the KG as $G$ and restrict the unary formulas in CML$[G, \mathsf{h}]$ to the form of $R(\mathsf{h}, x)$. This theorem is directly obtained by Theorem C.2 because constant $h$ can be equivalently transformed to constant predicate $P_h(x)$. $\square$

*Proof of Corollary 3.3.* **Base case:** Since the unary predicate can be encoded into the initial representation of the entity according to Section C.1. Then the base case is obvious.

**Recursion rule:** Since the rule structures $R(\mathsf{h}, x), R_1(\mathsf{h}, x), R_2(\mathsf{h}, x)$ are unary predicate and can be learned by QL-GNN, they are formulas in CML$[G, \mathsf{h}]$. According to recursive definition of CML, $R_1(\mathsf{h}, x) \wedge R_2(\mathsf{h}, y), \exists^{\geq N} y \, (R_i(y, x) \wedge R(\mathsf{h}, y))$ are also formulas in CML$[G, \mathsf{h}]$, therefore can be learned by QL-GNN.

$\square$

## C.4 Proof of Theorem 3.4

**Definition C.13.** CompGCN learns a rule formula $R(x, y)$ if and only if given any graph $G$, the QL-GNN's score of a new triplet $(h, R, t)$ can be mapped to a binary value, where `True` indicates that $R(x, y)$ satisfies on entity pair $(h, t)$, while `False` does not satisfy.

*Proof.* According to Theorem C.2, the MPNN representation $\mathbf{e}_v^{(L)}$ can represent the formulas in CML$[G]$. Assume $\varphi_1(x)$ and $\varphi_2(y)$ can be represented by the MPNN representation $\mathbf{e}_v^{(L)}, v \in \mathcal{V}$ and there exists two functions $g_1$ and $g_2$ that can extract the logical formulas from $\mathbf{e}_v^{(L)}$, i.e., $g_i(\mathbf{e}_v^{(L)}) = 1$ if $G, v \models \varphi_i$ and $g_i(\mathbf{e}_v^{(L)}) = 0$ if $G, v \nvDash \varphi_i$ for $i = 1, 2$. We show how the following two logical operators can be learned by $s(h, R, t)$ for candidate triplet $(h, R, t)$:

- Conjunction: $\varphi_1(x) \wedge \varphi_2(y)$. The conjunction of $\varphi_1(x), \varphi_2(y)$ can be learned with function $s(h, R, t) = g_1(\mathbf{e}_h^{(L)}) \cdot g_2(\mathbf{e}_t^{(L)})$.

- Negation: $\neg\varphi_1(x)$. The negation of $\varphi_1(x)$ can be learned with function $s(h, R, t) = 1 - g_1(\mathbf{e}_h^{(L)})$.

The disjunction $\vee$ can be obtained by $\neg(\neg\varphi_1(x) \wedge \neg\varphi_2(y))$. More complex formula involving sub-formulas from $\{\varphi(x)\}$ and $\{\varphi'(y)\}$ can be learned by combining the score functions above. $\qquad\square$

### C.5 PROOF OF PROPOSITION 4.1

**Lemma C.14.** *Assume $\varphi(x)$ describes a single-connected rule structure $\mathsf{G}$ in a KG. If assign constant to entities with out-degree large than 1 in the KG, the structure $\mathsf{G}$ can be described with formula $\varphi'(x)$ in CML of KG with assigned constants.*

*Proof.* According to Theorem C.7, assume $\varphi'(x)$ with assigned constants is not equivalent to a formula in CML, there should exist two rule structures $\mathsf{G}, \mathsf{G}'$ in KG $G, G'$, and entity $v$ in $\mathsf{G}$ and entity $v'$ in $\mathsf{G}'$ such that $\mathrm{Unr}_{\mathsf{G}}^L(v) \cong \mathrm{Unr}_{\mathsf{G}'}^L(v')$ for every $L \in \mathbb{N}$ and such that $\mathsf{G}, v \models \varphi'$ but $\mathsf{G}', v' \nvDash \varphi'$.

Since each entity in $\mathsf{G}$ ($\mathsf{G}'$) with out-degree larger than 1 is assigned with a constant, the rule structure $\mathsf{G}$ ($\mathsf{G}'$) can be uniquely recovered from its unravelling tree $\mathrm{Unr}_{\mathsf{G}}^L(v)$ ($\mathrm{Unr}_{\mathsf{G}}^L(v)$) for sufficient large $L$. Therefore, if $\mathrm{Unr}_{\mathsf{G}}^L(v) \cong \mathrm{Unr}_{\mathsf{G}'}^L(v')$ for every $L \in \mathbb{N}$, the corresponding rule structures $\mathsf{G}$ and $\mathsf{G}'$ should be isomorphism too, which means $\mathsf{G}, v \models \varphi'$ and $\mathsf{G}', v' \models \varphi'$. Thus, $\varphi'(x)$ must be a formula in CML. $\qquad\square$

*Proof of Proposition 4.1.* The theorem holds by restricting the unary formula to the form of $R(\mathsf{h}, x)$ on Lemma C.14. $\qquad\square$

*Proof of Corollary 4.2.* By converting new constants $\mathsf{c}_1, \mathsf{c}_2, \cdots, \mathsf{c}_k$ to constant predicates $P_{c_1}(x), P_{c_2}(x), \cdots, P_{c_k}(x)$, the corollary holds by using Theorem 3.2. $\qquad\square$

## D EXPERIMENTS

### D.1 MORE RULE STRUCTURES IN SYNTHETIC DATASETS

In Section 6.1, we also include the following rule structures in the synthetic datasets, i.e., $C_4$ and $I_2$ in Figure 6, for experiments. $C_4$ and $I_2$ are both formulas from $\mathrm{CML}[G, \mathsf{h}]$. The proof of $C_4$ is similar to the proof of $C_3$ in Corollary A.2. The proof of $I_2$ is similar to that of $I_1$ and is in Corollary D.1.

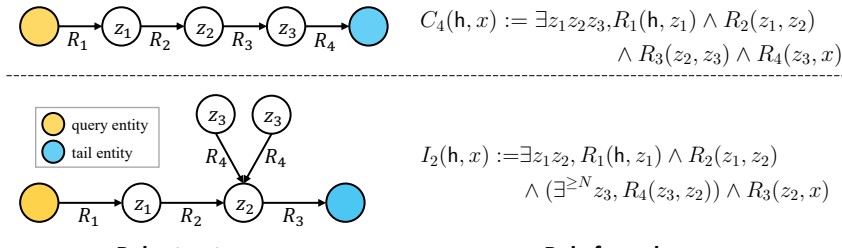

$$C_4(\mathsf{h}, x) := \exists z_1 z_2 z_3, R_1(\mathsf{h}, z_1) \wedge R_2(z_1, z_2) \wedge R_3(z_2, z_3) \wedge R_4(z_3, x)$$

$$I_2(\mathsf{h}, x) := \exists z_1 z_2, R_1(\mathsf{h}, z_1) \wedge R_2(z_1, z_2) \wedge (\exists^{\geq N} z_3, R_4(z_3, z_2)) \wedge R_3(z_2, x)$$

**Rule structure**          **Rule formula**

Figure 6: In the synthetic experiments, we also compare the performance of various GNNs on the synthetic datasets generated from $C_4$ and $I_2$.

**Corollary D.1.** *$I_2(\mathsf{h}, x)$ is a formula in CML$[G, \mathsf{h}]$.*

*Proof.* $I_2(\mathsf{h}, x)$ is a formula in $\mathrm{CML}[G, \mathsf{h}]$ as it can be recursively defined as follows

$$
\begin{aligned}
\varphi_1(x) &= P_h(x), \\
\varphi_2(x) &= \exists y, R_1(y, x) \wedge \varphi_1(y), \\
\varphi_3(x) &= \exists y, R_2(y, x) \wedge \varphi_2(y), \\
\varphi_s(x) &= \exists^{\geq 2} y, R_4(y, x) \wedge \top, \\
\varphi_4(x) &= \varphi_s(x) \wedge \varphi_3(x), \\
I_2(\mathsf{h}, x) &= \exists y, R_3(y, x) \wedge \varphi_4(y).
\end{aligned}
$$

$\square$

## D.2 EXPERIMENTS FOR COMPGCN

The classical framework of KG reasoning is inadequate for assessing the expressivity of CompGCN because the query $(h, R, ?)$ assumes that certain logical formula $\varphi(x)$ are satisfied at the head entity $h$ by default. In order to validate the expressivity of CompGCN, it is necessary to predict all missing triplets directly based on entity representations without relying on the query $(h, R, ?)$. To accomplish this, we create a new dataset called $S$ that adheres to the rule formula $S(x, y) = \varphi^\star(x) \wedge \varphi^\star(y)$, where the logical formula is defined as:

$$
\varphi^\star(x) = \exists y R_1(x, y) \wedge (\exists x R_2(y, x) \wedge (\exists y R_3(x, y))).
$$

Here, $\varphi^\star(x)$ is represented with parameter reusing (reusing $x$ and $y$) and is a formula in CML. Therefore, the formula $S(x, y)$ takes the form of $R(x, y) = f_R(\{\varphi(x)\}, \{\varphi'(y)\})$ and can be learned by CompGCN, as indicated by Theorem 3.4. To validate our theorem, we generate a synthetic dataset $S$ using the same steps outlined in Section 6.1, following the rule $S(x, y)$. We then train CompGCN on dataset $S$. The experimental results demonstrate that CompGCN effectively learns the rule formula $S(x, y)$ with 100% accuracy. Comparing it with QL-GNN is unnecessary since the latter is specifically designed for KG reasoning setting involving the query $(h, R, ?)$.

## D.3 STATISTICS OF SYNTHETIC DATASETS

Table 6: Statistics of the synthetic datasets.

| Datasets | $C_3$ | $C_4$ | $I_1$ | $I_2$ | $T$ | $U$ | $S$ |
|---|---|---|---|---|---|---|---|
| known triplets | 1514 | 2013 | 843 | 1546 | 2242 | 2840 | 320 |
| training | 1358 | 2265 | 304 | 674 | 83 | 396 | 583 |
| validation | 86 | 143 | 20 | 43 | 6 | 26 | 37 |
| testing | 254 | 424 | 57 | 126 | 15 | 183 | 109 |

## D.4 RESULTS ON SYNTHETIC WITH MISSING TRIPLETS

We randomly remove 5%, 10%, and 20% edges from synthetic datasets to test the robustness of QL-GNN and EL-GNN for rule structures learning. The results of QL-GNN and EL-GNN are shown in Table 7 and 8 respectively. The results show that the completeness of rule structure correlates strongly with the performance of QL-GNN and EL-GNN.

Table 7: The accuracy of QL-GNN on synthetic datasets with missing triplets.

| Triplet missing ratio | $C_3$ | $C_4$ | $I_1$ | $I_2$ | $T$ | $U$ |
|---|---|---|---|---|---|---|
| 5% | 0.899 | 0.866 | 0.760 | 0.783 | 0.556 | 0.329 |
| 10% | 0.837 | 0.718 | 0.667 | 0.685 | 0.133 | 0.279 |
| 20% | 0.523 | 0.465 | 0.532 | 0.468 | 0.111 | 0.162 |

Table 8: The accuracy of EL-GNN on synthetic datasets with missing triplets.

| Triplet missing ratio | $C_3$ | $C_4$ | $I_1$ | $I_2$ | $T$ | $U$ |
|---|---|---|---|---|---|---|
| 5% | 0.878 | 0.807 | 0.842 | 0.857 | 0.244 | 0.5 |
| 10% | 0.766 | 0.674 | 0.725 | 0.661 | 0.222 | 0.347 |
| 20% | 0.499 | 0.405 | 0.637 | 0.458 | 0.111 | 0.257 |

### D.5 MORE EXPERIMENTAL DETAILS ON REAL DATASETS

**MRR and Hit@10** Here we supplement MRR and Hit@10 of NBFNet and EL-NBFNet on real datasets in Table 9. The improvement of EL-NBFNet on MRR and Hit@10 is not as significant as that on Accuracy because the EL-NBFNet is designed for exactly learning rule formulas and only Accuracy can be guaranteed to be improved.

Table 9: MRR and Hit@10 of NBFNet and EL-NBFNet on real datasets.

| | Family | | Kinship | | UMLS | | WN18RR | | FB15k-237 | |
|---|---|---|---|---|---|---|---|---|---|---|
| | MRR | Hit@10 | MRR | Hit@10 | MRR | Hit@10 | MRR | Hit@10 | MRR | Hit@10 |
| NBFNet | 0.983 | 0.993 | 0.900 | 0.997 | 0.970 | 0.997 | 0.548 | 0.657 | 0.415 | 0.599 |
| EL-NBFNet | 0.990 | 0.991 | 0.905 | 0.996 | 0.975 | 0.993 | 0.562 | 0.669 | 0.424 | 0.607 |

**Different hyperparameters of** $d$ We have observed that a larger or smaller $d$ does not necessarily lead to better performance in Figure 4. For real datasets, we also observed similar phenomenon in Table 10. For real datasets, we uses $d = 5, 30, 100, 100, 300$ for Family, Kinship, UMLS, WN18RR, and FB15k-237, respectively.

Table 10: The accuracy of EL-NBFNet on UMLS with different $d$.

| $d = 0$ | $d = 50$ | $d = 100$ | $d = 150$ | NBFNet |
|---|---|---|---|---|
| 0.948 | 0.958 | 0.963 | 0.961 | 0.951 |

**Time cost of EL-NBFNet** In Table 11, we show the time cost of EL-NBFNet and NBFNet on real datasets. The time cost is measured by seconds of testing phase. The results show that EL-NBFNet is slightly slower than NBFNet. The reason is that EL-NBFNet needs to traverse all entities on KG to assign constants to entities with out-degree larger than degree threshold $d$.

Table 11: Time cost (seconds of testing) of EL-NBFNet on real datasets.

| Methods | Family | Kinship | UMLS | WN18RR | FB15k-237 |
|---|---|---|---|---|---|
| EL-NBFNet | 270.3 | 14.0 | 6.7 | 35.6 | 20.1 |
| NBFNet | 269.6 | 13.5 | 6.4 | 34.3 | 19.8 |

## E THEORY OF GNNS FOR SINGLE-RELATIONAL LINK PREDICTION

Our theory of KG reasoning can be easily extended to the single-relational link prediction. The following two corollaries are the extensions of Theorem 3.2 and Theorem 3.4 to the single-relational link prediction, respectively.

**Corollary E.1** (Theorem 3.2 on single-relational link prediction). *For single-relational link prediction, given a query* $(h, R, ?)$*, a rule formula* $R(\mathsf{h}, x)$ *is learned by QL-GNN if and only if* $R(\mathsf{h}, x)$ *is a formula in CML$[G, \mathsf{h}]$.*

**Corollary E.2** (Theorem 3.4 on single-relational link prediction). *For single-relational link prediction, CompGCN can learn the rule formula $R(x, y) = f_R\left(\{\varphi(x)\}, \{\varphi'(y)\}\right)$ where $f_R$ is a logical formula involving sub-formulas from $\{\varphi(x)\}$ and $\{\varphi'(y)\}$ which are the sets of formulas in $CML[G]$ that can be learned by GNN (1).*

Corollary E.1 and E.2 can be directly proven by restricting the logic of KG to single-relational graph, which means there is only one binary predicate in logic of graph.

# F   UNDERSTANDING GENERALIZATION BASED ON EXPRESSIVITY

## F.1   UNDERSTANDING EXPRESSIVITY VS. GENERALIZATION

In this section, we provide some insights on the relation between expressivity and generalization. Expressivity in deep learning pertains to a model's capacity to accurately represent information, whereas the ability of a model to achieve this level of expressivity depends on its generalization. Considering generalization requires not only contemplating the model design but also assessing whether the training algorithm can enable the model to achieve its expressivity. The experiments in this paper can also show this relation about expressivity and generalization from two perspective: (1) The experimental results of QL-GNN shows that its expressivity can be achieved with classical deep learning training strategies; (2) In the development of deep learning, a consensus is that more expressivity often leads to better generalization. The experimental results of EL-GNN verify this consensus.

In addition, our theory can provide some insights on model design with better generalization. Based on the constructive proof of Lemma C.3, if QL-GNN can learn a rule formula $R(\mathsf{h}, x)$ with $L$ recursive definition, QL-GNN can learn $R(\mathsf{h}, x)$ with layers and hidden dimensions no less than $L$. Assuming learning $r$ relations with QL-GNN and numbers of recursive definition for these relations are $L_1, L_2, \cdots, L_r$ respectively, QL-GNN can learn these relations with layers no more than $max_i L_i$ and hidden dimensions no more than $\sum L_i$. Since these bounds are nearly worst-case scenarios, both the dimensions and layers can be further optimized. Also, in the constructive proof of Lemma C.3, the aggregation function is summation, and it is difficult for mean and max/min aggregation function to capture sub-formula $\exists^{\geq N} y\left(R_i(y, x) \wedge R(\mathsf{h}, y)\right)$. From the perspective of rule learning, QL-GNN extracts structural information at each layer. Therefore, to learn rule structures, QL-GNN needs an activation function with compression capability for information extraction from inputs. Empirically, QL-GNN with identify activation function fails to learn with rules in synthetic dataset.

Moreover, because our theory cannot help understand generalization related to network training, the dependence to hyperparameters of network training, e.g., the number of training examples, graph size, number of entities, cannot be revealed from our theory.

## F.2   WHY ASSIGNING LOTS OF CONSTANTS HURTS GENERALIZATION?

We take the relation $C_3$ as an example to show why assigning lots of constants hurts generalization from logical perspective. We add two different constants $\mathsf{c}_1$ and $\mathsf{c}_2$ to the rule formula $C_3(h, x)$, which results two different rule formulas $C_3'(\mathsf{h}, y) = \exists z_1 R_1(\mathsf{h}, z_1) \wedge R_2(z_1, \mathsf{c}_1) \wedge R_3(\mathsf{c}_1, x)$ and $C_3^\star(\mathsf{h}, y) = \exists z_1 R_1(\mathsf{h}, z_1) \wedge R_2(z_1, \mathsf{c}_2) \wedge R_3(\mathsf{c}_2, x)$. Predicting new triplets for relation $C_3$ can now be achieved by learning the rule formulas $C_3(\mathsf{h}, x), C_3'(\mathsf{h}, x)$, or $C_3^\star(\mathsf{h}, x)$. Among these rule formulas, $C_3(\mathsf{h}, x)$ is the rule with the best generalization, while $C_3'(\mathsf{h}, x)$ and $C_3^\star(\mathsf{h}, x)$ require the rule structure to pass through the entities with identifiers of constants $\mathsf{c}_1$ and $\mathsf{c}_2$, respectively. Thus, when adding constants, maintaining performance requires the network to learn both rule formulas $C_3'(\mathsf{h}, x), C_3^\star(\mathsf{h}, x)$ simultaneously which may potentially require a network with larger capacity. Even EL-GNN is unnecessary to learn $C_3'(\mathsf{h}, x), C_3^\star(\mathsf{h}, x)$ since $C_3(\mathsf{h}, x)$ is learnable, EL-GNN cannot avoid learning rules with more than one constant in it when the rules are out of CML.

# G   LIMITATIONS AND IMPACTS

Our work offers a fresh perspective on understanding GNN's expressivity in KG reasoning. Unlike most existing studies that focus on distinguishing ability, we analyze GNN's expressivity based solely on its ability to learn rule structures. Our work has the potential to inspire further studies. For

instance, our theory analyzes GNN's ability to learn a single relation, but in practice, GNNs are often applied to learn multiple relations. Therefore, determining the number of relations that GNNs can effectively learn for KG reasoning remains an interesting problem that can help determine the size of GNNs. Furthermore, while our experiments are conducted on synthetic datasets without missing triplets, real datasets are incomplete (e.g., missing triplets in testing sets). Thus, understanding the expressivity of GNNs for KG reasoning on incomplete datasets remains an important challenge.

