# OpenReview forum: "Understanding Expressivity of GNN in Rule Learning"
_ICLR.cc/2024/Conference — ICLR 2024 poster_

### Official Review · Reviewer_TfWh · 2023-10-30

**Soundness:** 3 good
**Presentation:** 3 good
**Contribution:** 3 good
**Rating:** 8
**Confidence:** 3

**Summary:**

In this paper, the authors investigate the expressivity of GNN-based KG reasoning methods. The paper provides terminology for rule structure in the KG reasoning task and proves the limitation of current methods in structure rules T(h,x) and U(h,x) in theory. For the method, In algo 1, the initial representation is assigned to the entities whose out-degree is larger than a threshold d. The proposition 5.1 and the empirical results in Table 1 show that the proposed method can help discover structure rules T(h,x) and U(h,x).

**Strengths:**

- The novelty is great, this paper provides a systematic way for rule structures finding in GNN based KG reasoning task.
- The theoretical part is sound, and the experimental study supports the theoretical result as well.
- The method is simple that assigning the initial representation to the entities but effective based on experimental results.

**Weaknesses:**

n/a

**Questions:**

In page 7, it mentioned " the additional time complexity introduced by entity labeling is linear with respect to the number of entities in the graph,  which is marginal in comparison to QL-GNN". I am not sure about why is marginal considering the KG is usually large. Could you provide the numeric value about time cost for the experiments or adding more discussion?

---

> ### Author Response · Authors · 2023-11-21
> **Response to Reviewer TfWh**
>
> Thank you for taking the time to review our work and provide valuable feedbacks.
>
> ### Q. I am not sure about why is marginal considering the KG is usually large. Could you provide the numeric value about time cost for the experiments or adding more discussion?
>
> **A.**
> Thank you for your feedback.
> Despite KG being large, the additional time required for EL-GNN is insignificant compared to QL-GNN. This is because the extra time in EL-GNN arises from traversing all entities on KG, which is not the main factor affecting time cost. The primary contributor to time cost is message passing and other time-consuming operations in QL-GNN.
> To make the time comparison clearer, we improve the writing in section 4 of revision.
> Also, we compare the time costs (seconds of testing) of QL-GNN(NBFNet) and EL-GNN(EL-NBFNet) in various KGs in the following table (Table 11 in revision).
>
> | Methods | Family | Kinship | UMLS | WN18RR | FB15k-237 |
> |---------|--------|---------|------|--------|-----------|
> |EL-GNN|270.3   |14.0     |6.7   |35.6    |20.1       |
> |QL-GNN   |269.6   |13.5     |6.4   |34.3    |19.8       |
>
> The time costs of experiments are consistent with our analysis in Section 4 of revision and indicate a negligible additional cost(~1s) for EL-GNN compared to QL-GNN.

---

### Official Review · Reviewer_S7Jh · 2023-10-31

**Soundness:** 3 good
**Presentation:** 3 good
**Contribution:** 3 good
**Rating:** 6
**Confidence:** 3

**Summary:**

The paper delves into the domain of Knowledge Graph (KG) reasoning, which involves deducing new facts from existing ones in a KG. While Graph Neural Networks (GNNs) with tail entity scoring have recently achieved state-of-the-art performance in KG reasoning, there's a gap in the theoretical understanding of these GNNs. This work aims to bridge this gap by unifying GNNs with tail entity scoring into a common framework and analyzing their expressivity in terms of the rule structures they can learn. The insights from this analysis lead to the proposal of a novel labeling strategy to further enhance rule structure learning in KG reasoning. Experimental results support the theoretical findings and demonstrate the effectiveness of the proposed method.

**Strengths:**

S1 The paper provides a thorough analysis of the expressivity of state-of-the-art GNNs used for KG reasoning. By unifying these GNNs into a common framework (QL-GNN), the authors offer a structured approach to understanding their capabilities and limitations in terms of rule structure learning.
S2 The introduction of the QL-GNN framework and the subsequent EL-GNN model showcases the authors' innovative approach to addressing the gaps in the current understanding of GNNs for KG reasoning. The EL-GNN, in particular, is designed to learn rule structures beyond the capacity of existing methods, marking a significant advancement in the field.
S3 The authors don't just rely on theoretical findings; they validate their claims with experiments on synthetic datasets. The consistency between the experimental results and theoretical insights adds credibility to their claims and demonstrates the practical applicability of their proposed methods.

**Weaknesses:**

W1  While the EL-GNN model is introduced as an improvement over QL-GNN, there's limited discussion on its scalability. How does EL-GNN perform when applied to very large-scale KGs? Are there any computational constraints or challenges that users should be aware of? Moreover, the experiments in the paper employ relatively small datasets. It would greatly benefit the research to include larger datasets to demonstrate the effectiveness of the proposed methods on a more substantial scale.

W2 The paper introduces a novel labeling strategy to enhance rule structure learning. Are there specific scenarios in which this labeling strategy may not yield effective results or encounter limitations?

**Questions:**

Q1 Can the models trained on one KG be adapted or fine-tuned for another KG? If so, are there any specific considerations or challenges in doing so?

Q2 How robust are QL-GNN and EL-GNN to noisy or incomplete data in KGs? Have any tests been conducted to assess their performance under such conditions?

---

> ### Author Response · Authors · 2023-11-21
> **Response to Reviewer S7Jh**
>
> Thank you for taking the time to review our work and provide valuable feedbacks.
>
> ### W1. Experiments
>
> **W1.1** How does EL-GNN perform when applied to very large-scale KGs? Are there any computational constraints or challenges that users should be aware of?
>
> **R1.1** For large-scale KGs, EL-GNN has a similar scalability with QL-GNN, because the extra time cost of EL-GNN comes from traversing all entities on KG, which costs negligibly compared to QL-GNN. We compare the time cost (seconds of testing) of QL-GNN(NBFNet) and EL-GNN(EL-NBFNet) on different KGs in the following table (in Table 11 of revision).
> The results show that EL-GNN has negligible extra cost (~1s) when compared to QL-GNN.
>
> | Methods | Family | Kinship | UMLS | WN18RR | FB15k-237 |
> |---------|--------|---------|------|--------|-----------|
> |EL-GNN   |270.3   |14.0     |6.7   |35.6    |20.1       |
> |QL-GNN   |269.6   |13.5     |6.4   |34.3    |19.8       |
>
> **W1.2** Experiments on large datasets
>
> **R1.2** Thank you for the suggestion. In the revision, we conducted experiments on FB15K-237, a dataset with more edges. The accuracies of QL-GNN and EL-GNN are shown in the following table.
> In the following table, the accuracy of EL-GNN is higher than QL-GNN, which indicates that EL-GNN improves the performance on large-scale datasets.
>
> | EL-NBFNet | EL-RED-GNN | NBFNet | RED-GNN |
> |-----------| -----------|--------|---------|
> |0.332 ($\uparrow$ 3.5\%)      | 0.322($\uparrow$ 10\%)      | 0.321  | 0.284   |
>
> ### W2. Scenarios in which this labeling strategy may not yield effective results or encounter limitations
>
> **R2.** One possible limitation of EL-GNN is the risk of over-fitting due to its larger number of parameters compared to QL-GNN. However, by carefully choosing a suitable degree threshold d, a balance can be achieved between the fitting ability and performance of EL-GNN.

---

> > ### Author Response · Authors · 2023-11-21
> >
> > ### Q1. Can the models trained on one KG be adapted or fine-tuned for another KG? If so, are there any specific considerations or challenges in doing so?
> >
> > **A1.** The adaption of one KG to another is out of the scope of our paper. However, for such an important problem, our theory can indeed provide some insights.
> >
> > From the perspective of rule structure learning, we think that the adaption of one KG to another is possible if these two KGs share some common relations.
> > Assuming we want to transfer the model from KG `A` to KG `B`, one possible challenge is that the model has learned all the rule structures in `A`, but some of these rule structures are useless in KG `B`. Therefore, whether the model can forget these rules from `A` during the transfer process is a meaningful question, otherwise, it may lead to a capacity bottleneck for the model.
> >
> > ### Q2. How robust are QL-GNN and EL-GNN to noisy or incomplete data in KGs? Have any tests been conducted to assess their performance under such conditions?
> >
> > **A2.** Thank you for your questions. For the real-world KGs evaluated in Section 6.2, noise and incomplete data already exists there. The superior performance of EL-GNN and QL-GNN in Table 2 indicates their robustness.
> >
> > For the synthetic datasets, the experiments in Section 6.1 (Table 1) have taken noisy cases into consideration by inserting noisy triplets in each dataset.
> > The noisy ratio of each dataset is shown in the following table.
> >
> > |C3|C4|I1|I2|T|U|
> > |---|---|---|---|---|---|
> > |50%|50%|50%|57%|75%|67%|
> >
> > To further demonstrate the robustness of EL-GNN and QL-GNN, we conducted experiments on the synthetic dataset with more noisy triplets. The results are shown in the following tables.
> >
> > QL-GNN
> >
> > |noisy triplets ratio|C3|C4|I1|I2|T|U|
> > |---|---|---|---|---|---|---|
> > |70%|1.0|1.0|1.0|1.0|1.0|0.526|
> > |80%|1.0|1.0|1.0|1.0|1.0|0.481|
> > |90%|1.0|1.0|1.0|1.0|1.0|0.463|
> >
> > EL-GNN
> >
> > |noisy triplets ratio|C3|C4|I1|I2|T|U|
> > |---|---|---|---|---|---|---|
> > |70%|1.0|1.0|1.0|1.0|1.0|0.831|
> > |80%|1.0|1.0|1.0|1.0|1.0|0.797|
> > |90%|1.0|1.0|1.0|1.0|1.0|0.773|
> >
> > Therefore, the results in the above table demonstrate the robustness of EL-GNN and QL-GNN regarding noisy data.
> >
> > As for incomplete data, we randomly remove some triplets from the synthetic datasets, and report the results in the following tables(Table 7 and 8 in revision).
> >
> > QL-GNN
> >
> > |    removed triplets ratio        |     C3      |     C4      |     I1      |     I2      |     T        |     U        |
> > |------------|--------------|--------------|--------------|--------------|--------------|--------------|
> > |     5%     |     0.899    |     0.866    |     0.760    |     0.783    |     0.556    |     0.329    |
> > |     10%    |     0.837    |     0.718    |     0.667    |     0.685    |     0.133    |     0.279    |
> > |     20%    |     0.523    |     0.465    |     0.532    |     0.468    |     0.111    |     0.162    |
> >
> > EL-GNN
> >
> > |     removed triplets ratio       |     C3      |     C4      |     I1      |     I2      |     T        |     U        |
> > |------------|--------------|--------------|--------------|--------------|--------------|--------------|
> > |     5%     |     0.878    |     0.807    |     0.842    |     0.857    |     0.244    |     0.500      |
> > |     10%    |     0.766    |     0.674    |     0.725    |     0.661    |     0.222    |     0.347    |
> > |     20%    |     0.499    |     0.405    |     0.637    |     0.458    |     0.111    |     0.257    |
> >
> > As can be seen, the incomplete graph will result in a performance decline because the missing triplets produce some corrupted rule structures in KGs.

---

> > > ### Comment · Reviewer_S7Jh · 2023-11-22
> > > **Response to the authors**
> > >
> > > I appreciate the thorough responses. My opinion of the paper remains unchanged.

---

### Official Review · Reviewer_zRzb · 2023-10-31

**Soundness:** 3 good
**Presentation:** 3 good
**Contribution:** 3 good
**Rating:** 6
**Confidence:** 3

**Summary:**

This paper proposes a novel perspective to understand the expressivity of recent GNNs for KG reasoning based on rule structure learning. It identifies the types of rule structures that different GNNs can learn and analyzes their advantages and limitations. It also introduces a unified framework, QL-GNN, that encompasses two SOTA GNNs, RED-GNN and NBFNet. Moreover, it presents a new labeling strategy based on QL-GNN, called EL-GNN, that can learn more rule structures. The paper validates the theoretical analysis and the effectiveness of QL-GNN and EL-GNN through experiments on synthetic and real datasets.

**Strengths:**

- The paper presents a novel approach to understanding RED-GNN and NBFNet from a rule-learning perspective.

- The theoretical analysis reveals the advantages and limitations of existing popular GNNs in KG reasoning. The paper also provides experimental results to support the theoretical conclusion.

- Furthermore, the paper proposes two GNNs for KG reasoning, which outperform state-of-the-art models on several datasets.

**Weaknesses:**

- In my opinion, the datasets used in Section 6.2 appear to be well-suited for rule-based methods. However, the most popular link prediction dataset, FB15K-237, was not included in this experiment. Therefore, I believe that the experimental results of Section 6.2 are insufficient to evaluate the effectiveness of the proposed method.

**Questions:**

- What about the performance of the proposed GNNs on FB15K-237?

---

> ### Author Response · Authors · 2023-11-21
> **Response to Reviewer zRzb**
>
> Thank you for taking the time to review our work and provide valuable feedbacks.
>
> ### W1 & Q1. Performance on FB15K-237
>
> **R1.** Thank you for your suggestions. To fully show the effectiveness of EL-GNN, we conducted experiments on FB15k-235 and added the results to Table 2 of revision. Below is the accuracy of EL-GNN and QL-GNN on FB15k-237.
> From this table, we can see that EL-GNN outperforms QL-GNN on FB15k-237, indicating that EL-GNN can improve performance on large-scale datasets.
>
>
> | EL-NBFNet              | EL-RED-GNN | NBFNet | RED-GNN |
> | ---------------------- | ---------- | ------ | ------- |
> | 0.332 ($\uparrow$ 3.5\%) | 0.322($\uparrow$ 10\%)      | 0.321  | 0.284   |

---

### Official Review · Reviewer_NwzR · 2023-10-31

**Soundness:** 3 good
**Presentation:** 2 fair
**Contribution:** 2 fair
**Rating:** 5
**Confidence:** 3

**Summary:**

This paper which studies theoretical properties of GNN models includes:
* Analysis of the expressiveness of GNN models in terms of a rule-learning formalism.
* Presentation a simple yet effective labeling strategy based on their analysis that yields improvements.
* Empirical analysis that supplements the theoretical contribution. The proposed approach instantiated with RED-GNN and NBFNet yields positive improvements across real and synthetic datasets.

**Strengths:**

This paper leads a technical and thoughtful analysis to what kinds of relationships GNN-based models can effectively represent and effectively predict. Strengths of the paper include:
* **Formalism** for link prediction in knowledge graphs using CML. This allows the authors to describe which kinds of rule structures each class of model is able to represent. It allows for the generalization of existing methods to represent broader classes of rules.
* **Empirical Successes** are demonstrated across a wide variety of datasets. These seem to indicate different kinds of graph and entity / relation structures.
* **Theoretical Analysis** appears to be rigorous and formally describe the different classes of rules and what models are effective for each.

**Weaknesses:**

My main concern with this paper is the presentation / structure and the way in which that presentation and structure limits the reader from connecting both the clear theoretical advantages of the proposed class of GNN to both the limitations of other classes and the empirical successes. Please correct me if you think I have misinterpreted or misunderstood things or put emphasis points inappropriately. I am mentioning these presentation points because I think the paper has a number of very nice properties that I would like readers to be able to more easily grasp and benefit from.

W1. **Defining Expressivity** I think that the definition of expressivity used in the paper should be defined much earlier in the manuscript. I say this not only for the sake of readers unfamiliar with definitions of expressivity for GNNs but also for the sake of familiar readers understanding differences between the choice of expressivity definition and the choices of past work. The related work section, which appears before definitions of expressivity so far as I understand, provides too high level a comparison between past work to be meaningful (in my opinion) for all but the most familiar readers of these methods (as an aside, I think that the related work section as it is now would be better suited later in the paper, say before experiments). In my opinion, readers would benefit from explicitly talking about the relationship between generalization and representation of graphs immediately.

W2. **Connecting Formalism and Data** While the formalism used is based on past work and as I understand motivated and accepted in those works as a meaningful formalism, I think that paper would be greatly improved with many more motivating examples from real data that express why the rule based formalism is meaningful. For instance, the example in Figure 1 is great. I see how it connects to Figure 3 and Corollary 5.2. However, how often do such patterns appear in the real world empirical datasets? How much of the gains from the given methods correlate with the existence of the kinds of subgraphs described?

W3. **Understanding Generalization** I think my main point of confusion, which I was not able to resolve in my reading of the paper is how to think about generalization vs representation. As I understand it depends on which rules the model can learn. But I am having a hard time understanding how this relates to things like number of parameters, number of training examples, choice of aggregation functions, graph size, number of entities, number of relationships, etc. I am missing something fundamental here? E.g. How do number of examples / graph size / parameters relate to the base theorem C.2? Or do those things not matter in the analysis? It seems it depends only on $L$ is that correct?

W4. **Presentation of some of the theoretical results** As a more minor point, I think readers would take away more from the theoretical results if the authors provided more remarks about the limitations/take aways from the theorem statements. For instance, I was confused about Theorem 4.4. For instance, I think I would have appreciated more handholding as to the result: "The structural rules in Figure 2 cannot be learned by CompGCN due to Theorem 4.4."

My concern is that I think we need:
 (1) how the proposed formalism allows us to better analyze the generalization capabilities of models to understand why we would expect empirical successes
 (2) how the proposed formalism is reflected in real world datasets (e.g., the kinds of rule patterns indeed show up)
 (3) an understanding of why the proposed formalism and analysis is better than other forms of analysis that one could do.
These are certainly addressed by the paper, but I think that they could made significantly more crisp in the way the paper and results are presented.

Minor:
* I think the first sentence is missing an "A", "A knowledge graph (KG) ..."

Other related work:
* [Ego-Splitting Framework: from Non-Overlapping to Overlapping Clusters.](https://dl.acm.org/doi/10.1145/3097983.3098054)
* [Neighborhood Growth Determines Geometric Priors for Relational Representation Learning](https://proceedings.mlr.press/v108/weber20a.html)
* [What relations are reliably embeddable in Euclidean space?](https://arxiv.org/abs/1903.05347)

**Questions:**

* Can you say more about how to think about generalization and expressivity in regards to the above comments?
* Can you say more about Theorem 4.4 and "The structural rules in Figure 2 cannot be learned by CompGCN due to Theorem 4.4."?

---

> ### Author Response · Authors · 2023-11-21
> **Response to Reviewer NwzR**
>
> Thank you for taking the time to review our work and provide constructive feedbacks. We have followed your suggestion to improve the structures and presentations of our paper.
>
> First, we emphasize the significance of expressivity for model's generalization. Expressivity refers to a model's ability to represent information, while generalization is the expressivity a model can reach in practice after thorough training process. The experiments in this paper show that QL-GNN can reach such expressivity with standard deep learning training methods. On the other hand, a consensus in deep learning is that more expressivity typically leads to better generalization(e.g., GNN[1] & Transformer[2]). Our experiments also demonstrate that EL-GNN, with stronger expressivity, indeed leads to stronger generalization. The generalization of GNN is also the topic we are most interested in. We are glad to have more discussion with you.
>
> ### W1. Defining Expressivity.
> **W1.1** Definition of expressivity used in the paper should be defined much earlier.
>
> **R1.1** The expressivity in our paper use logic as tools to study the ability of GNN for learning rule structures rather than distinguishing structures with different triplet representations, such as [3, 4].
> Actually, the expressivity in our paper is called ``logical expressivity`` in related fields. To ease your concern and make the definition clearer, we have made the following revisions:
>
> - we used the term ``logical expressivity`` to emphasize the difference of expressivity in our paper from previous work;
>
> - we added a few sentences in Section 3.1 to explain what ``logical expressivity`` is (i.e., logical expressivity of GNN is a measurement of the ability of GNN to learn logical formulas and is defined as the set of logical formulas that GNN can learn);
>
> - we revised the connection between logical expressivity and learning rule structures in Section 3.2.
>
> **W1.2** related work section has high level comparison & discuss about generalization and representation immediately
>
> **R1.2** Thank you for the suggestion. The related work has been moved to the place before the experiment section in the latest revision.
> In addition, we emphasized the relationship between generalization and expressivity in the 3rd paragraph of the introduction section in revision by pointing out that generalization is the maximum expressivity that QL-GNN can generalize through training.

---

> > ### Author Response · Authors · 2023-11-21
> >
> > ### W2. Connecting Formalism and Data
> >
> > **W2.1** Many more motivating examples from real data that express why the rule based formalism is meaningful
> >
> > **R2.1** Following your suggestion, we provide additional motivating rule structures from Family and FB15k-237, the real-world datasets we conducted experiments on, in Figure 5 (Appendix A) in the revision. The examples in Figure 5(b) and 5(c) can be proved to be CML with Corollary 3.3 while Figure 5(a) cannot, which shows that rules defined by CML are common in real-world datasets and the rules beyond CML also exist.
> >
> > **W2.2** How often do such patterns appear in real-world empirical datasets?
> >
> > **R2.2** Simple rule patterns (e.g., chain-like rules) often have high occurrence in the real world. For example, rule-based methods such as Neural LP [7] and DRUM [8] utilize chain-like rule structures for KG reasoning and their performances indicate that KGs contain numerous chain-like rules.
> > Apart from simple rule patterns, there also exist many complex patterns in KG [5,6,9,10].
> > For complex rule patterns, counting their frequency is currently intractable due to the lack of tools to accurately detect complex rule structures in KG. However, the performance of QL-GNN improves a lot over rule-based methods in all the given datasets, which indicates that there exists more complex rule patterns in KGs.
> >
> > Furthermore, it is important to note that the success of QL-GNN does not stem from the prevalence of a particular rule pattern in KG. Rather, it is attributed to KG's high occurrence of the class of rule patterns defined by CML. Referring to the visualized rule structures in NBFNet[5] and A*Net[6], we can observe that their interpretable rule structures fall within the category of rules defined by CML. This suggests that the rules defined by CML frequently occur in KG, thus elucidating why QL-GNN is effective for reasoning on KGs.
> >
> >
> > **W2.3** How much of the gains from the given methods correlate with the existence of the kinds of subgraphs described?
> >
> > **R2.3** Considering the existence of a specific rule structure in real datasets is hard to determine, we are unable to answer this question based on real datasets. Instead, we conducted an experiment on the synthetic dataset with incomplete rule structures. Specifically, we randomly removed some triplets from the synthetic dataset to corrupt the subgraph structure of rules and re-evaluated the performance of QL-GNN and EL-GNN on these datasets as follows.
> >
> > QL-GNN
> >
> > |   ratio of incomplete subgraph    |     C3      |     C4      |     I1      |     I2      |     T        |     U        |
> > |------------|--------------|--------------|--------------|--------------|--------------|--------------|
> > |     5%     |     0.891    |     0.862    |     0.753    |     0.775    |     0.551    |     0.323    |
> > |     10%    |     0.832    |     0.710    |     0.659    |     0.681    |     0.128    |     0.273    |
> > |     20%    |     0.521    |     0.460    |     0.527    |     0.463    |     0.107    |     0.158    |
> >
> > EL-GNN
> >
> > |    ratio of incomplete subgraph        |     C3      |     C4      |     I1      |     I2      |     T        |     U        |
> > |------------|--------------|--------------|--------------|--------------|--------------|--------------|
> > |     5%     |     0.873    |     0.801    |     0.836    |     0.847    |     0.241    |     0.491      |
> > |     10%    |     0.763    |     0.671    |     0.722    |     0.656    |     0.220    |     0.343    |
> > |     20%    |     0.492    |     0.401    |     0.632    |     0.453    |     0.106    |     0.253    |
> >
> > As can be seen, the incomplete graph will result in a performance decline because the missing edges produce some corrupted rule structures in KG, indicating the gains from the given methods correlate strongly with the existence of the rule structures. We hope that these results on incomplete synthetic datasets can answer your question.

---

> > > ### Author Response · Authors · 2023-11-21
> > >
> > > ### W3. Understanding Generalization
> > >
> > >
> > > **W3.1** How to think about generalization vs representation
> > >
> > > **R3.1** First, it is important to clarify that generalization and expressivity are distinct concepts. Expressivity pertains to a model's capacity to accurately represent information, whereas the ability of a model to achieve this level of expressivity depends on its generalization.
> > > Considering generalization requires not only contemplating the model design but also assessing whether the training algorithm can enable the model to achieve its expressivity.
> > > The experiments in our paper also show this relation about expressivity vs generalization from two perspectives:
> > >
> > > - The experimental results of QL-GNN show that its expressivity can be achieved with classical training strategies for deep learning.
> > > - In the development of deep learning, a consensus is that more expressivity often leads to better generalization. The experimental results of EL-GNN verify this consensus.
> > >
> > > **W3.2** Understanding how generalization relates to things like number of parameters, number of training examples, choice of aggregation functions, graph size, number of entities, number of relationships, etc.
> > >
> > > **R3.2** Generalization relates to both the expressivity and training strategy of deep network. Based on the theoretical understanding of GNN's expressivity, we can provide some insights on designing a model with better generalization:
> > >
> > > - Layer & hidden dimension: if QL-GNN can learn a rule formula $R(\mathsf{h}, x)$ with $L$ recursive definition, QL-GNN can learn $R(\mathsf{h}, x)$ with layers and hidden dimensions no more than $L$.
> > > - Number of relations: Assume the number of relations is $r$ and the numbers of recursive definition for these relations are $L_1, L_2, \cdots, L_r$, QL-GNN can learn these relations with layers no more than $max_i L_i$ and hidden dimensions no more than $\sum_i L_i$.
> > > - Aggregation function: sum aggregation can learn the sub-formula $\exists^{\geq N}y \left( R_i(y, x) \wedge R(\mathsf{h}, y) \right)$, while max/mean cannot.
> > > - Activation function: activation function that can extract information from inputs (i.e., compress information) can learn rule structures well. Empirically, identity activation function fails to learn rule structures.
> > >
> > > The above concepts are related to the model design of QL-GNN (refer to Appendix F) and show the potential benefits of our theory in designing a model with better generalization. We also add the discussion about generalization in Appendix F of revision.
> > >
> > >
> > > ### W4. Presentation of some of the theoretical results, e.g., Theorem 4.4
> > >
> > > **R4.** We further discussed limitation and insights about generalization of Theorem 3.1 in the paragraph below Corollary 3.3 (i.e., limitation on learning single relation and insights on designing a model with better generalization) and added a remark to make Theorem 4.4 (3.4 in revision) clearer in revision.
> > >
> > > The reason for "The structural rules in Figure 2 cannot be learned by CompGCN" is that CompGCN encodes structures in the query and tail entities independently and fails to learn connected rule structures.

---

> > > > ### Author Response · Authors · 2023-11-21
> > > >
> > > > ### Q1. Say more about how to think about generalization and expressivity in regards to the above comments
> > > >
> > > > **A1.** Thank you for your suggestion. A more detailed discussion of generalization can be referred to in the response of W3. Here, we summarize some key points.
> > > >
> > > > Expressivity represents the ability of a model architecture to represent some information, while generalization refers to the ability of a model to achieve expressivity based on certain training strategies. Therefore, expressivity is an upper bound that a model can generalize to.
> > > > The experiments in this paper demonstrate that QL-GNN can reach this upper bound based on existing training methods.
> > > > On the other hand, from the development of deep learning, stronger expressivity generally leads to stronger generalization.
> > > > The experiments in this paper also demonstrate that GNN with stronger expressivity actually leads to stronger generalization in EL-GNN.
> > > >
> > > > ### Q2. More about Theorem 4.4
> > > >
> > > > **A2.** Thank you for your question, you can also refer to the response of W4 for a short discussion of this question.
> > > > Theorem 4.4 indicates that representations of two end entities encode two formulas respectively, and the structures described by these two formulas are independent (there is no guarantee they are connected). Thus, the rule structures learned by CompGCN should be two disconnected subgraphs surrounding the query and tail entities respectively.
> > > >
> > > > [1] How powerful are graph neural networks? ICLR 2019
> > > >
> > > > [2] Bert: Pre-training of deep bidirectional transformers for language understanding. ACL 2019
> > > >
> > > > [3] On the equivalence between positional node embeddings and structural graph representations. ICLR 2020
> > > >
> > > > [4] Labeling trick: A theory of using graph neural networks for multi-node representation learning. NeurIPS 2021
> > > >
> > > > [5] Neural bellman-ford networks: A general graph neural network framework for link prediction.NeurIPS 2021
> > > >
> > > > [6] A* net: A scalable path-based reasoning approach for knowledge graphs. NeurIPS 2023
> > > >
> > > > [7] Differentiable learning of logical rules for knowledge base reasoning. NeurIPS 2017
> > > >
> > > > [8] Drum: End-to-end differentiable rule mining on knowledge graphs. NeurIPS 2019
> > > >
> > > > [9] Embedding Logical Queries on Knowledge Graphs. NeurIPS 2018
> > > >
> > > > [10] Query2box: Reasoning over Knowledge Graphs in Vector Space using Box Embeddings. ICLR 2020

---

> > > > > ### Comment · Reviewer_NwzR · 2023-11-23
> > > > > **Thank you for your detailed reply**
> > > > >
> > > > > Thank you authors for your detailed reply. I believe that the changes strengthen the paper. I have, as a result, modified my review score from 3->5. I would encourage the authors to think more about the presentation of generalization & expressivity; I think there is room to make the points discussed here and in the revised paper even more crisp.

---

> > > > > > ### Author Response · Authors · 2023-11-23
> > > > > >
> > > > > > Thank you for your efforts and suggestions in helping us improve our work. We will continue to enhance the presentation of generalization & expressivity and make our paper more crisp.

---

### Author Response · Authors · 2023-11-23
**Summary of Revision**

We thank all reviewers for their valuable feedbacks. In summary, we have made the following revisions to the paper during the rebuttal period:

- We reorganized the paper structure to make it more readable (Section 5), and made the choice of expressivity clearer in Section 3.1. (Concerns of Reviewer NwzR)
- We discussed the relations between generalization and expressivity revealed by our theory in Appendix F. (Concerns of Reviewer NwzR)
- We conducted experiments on large-scale KG (FB15k-237) and added the results in Table 2. (Concerns of Reviewer zRzb and S7Jh)
- We compared the time costs of QL-GNN and EL-GNN in various KGs and added the results in Table 11. (Concerns of Reviewer S7Jh and TfWh)

We appreciate the reviewers' comments and suggestions and are open to further discussions.

---

### Meta-Review · Area_Chair_uonx · 2023-12-06

**Metareview:**

This work proposes a theoretical analysis of GNN methods for KG reasoning, mapping the rule structures learnable by MPNN with typed edges to rules for graded modal logic. The work describes which kinds of rule structures each class of model is able to represent. It allows for the generalization of existing methods to represent broader classes of rules.

- Reviewers were generally positive about the work. The rebuttal period saw fruitful discussion and one reviewer raising their score.

- Reading the theorems and proofs in the paper, at its core the work is translates the works of Barceló et al. to KG reasoning. There are some challenges in doing so, which justifies the proposed work.

**Justification For Why Not Higher Score:**

- At its core the work is translates the works of Barceló et al. to KG reasoning. There are some challenges in doing so, hence my recommendation of acceptance. I think it is a good, interesting analysis.

**Justification For Why Not Lower Score:**

- The KG community is not as familiar with notions of expresiveness as the geometric deep learning community. I think the reviewer scores are lower than they should be.

---

### Decision · Program_Chairs · 2024-01-16

Accept (poster)